# SaFT: Spotting Style Imitation and Filtering Content Interference for Zero-Shot LLM-Generated Text Detection

## Abstract

Large language models (LLMs) have achieved advanced text generation capabilities, necessitating the development of reliable LLM-generated text detection to prevent potential misuse. However, current probability-based zero-shot detection methods face two critical challenges that reduce the detection accuracy of LLM-generated texts: the *style imitation challenge (SIC)* and the *content interference challenge (CIC)*. The SIC arises as LLMs develop increasingly stronger abilities to mimic human writing styles, while the CIC occurs when surprising content characteristics interfere with probability analysis. To address these challenges, we propose *SaFT* [1], a novel framework built upon *Style-Oriented Instruction Prefix (SOIP)* to guide probability analysis for spotting style imitation and filtering content interference. Our framework proposes *SIC-Detection (SIC-D)* that spots style imitation by making style-imitating texts less unexpected through probability analysis conditioned on human-style instructions, and *CIC-Detection (CIC-D)* that filters content interference by difference analysis between probability distributions conditioned on contrasting style instructions, exploiting the insight that identical models exhibit equivalent content-related surprises. The final detection score is composed of SIC-D and CIC-D components. Extensive experiments demonstrate that SaFT consistently outperforms existing state-of-the-art methods, achieving improvements of 4.9% in average AUROC and 20.4% in average TPR @ 10% FPR.

## 1 Introduction

Due to continuously advancing next-token prediction methodologies, advanced large language models (LLMs) (OpenAI, 2024; Comanici et al., 2025) generate text with human-like authenticity that is virtually indistinguishable from human-written text. While these advanced capabilities have boosted productivity across numerous domains including academic research and journalism (M Alshater, 2022; Jiang et al., 2025), they have simultaneously introduced substantial risks through malicious applications such as academic dishonesty and fake news dissemination (Meyer et al., 2023; Deng et al., 2025), making robust and reliable LLM-generated text detection critically essential.

Existing detection research predominantly falls into embedding-based and probability-based methods. Embedding-based methods typically operate in a supervised manner, achieving strong performance by fine-tuning models like RoBERTa (Wang et al., 2023; Guo et al., 2024) on labeled datasets to learn discriminative representations. However, they suffer from domain-specific overfitting and poor cross-model generalization, often failing with unfamiliar domains or different LLM architectures (Ghosal et al., 2023; Wu et al., 2025). In contrast, probability-based detection operates in a zero-shot manner without requiring training data, employing pre-trained language models as scoring models to extract statistical metrics, including token likelihood scores (Solaiman et al., 2019), entropy measures (Gehrmann et al., 2019), and perturbation-based scoring mechanisms (Mitchell et al., 2023; Bao et al., 2024) that analyze inherent statistical distributions within text sequences. The underlying premise is that human-written text typically exhibits lower probability scores (more surprising to scoring models) compared to LLM-generated text, resulting in separable probability

---

[1]Our code and data will be released soon.

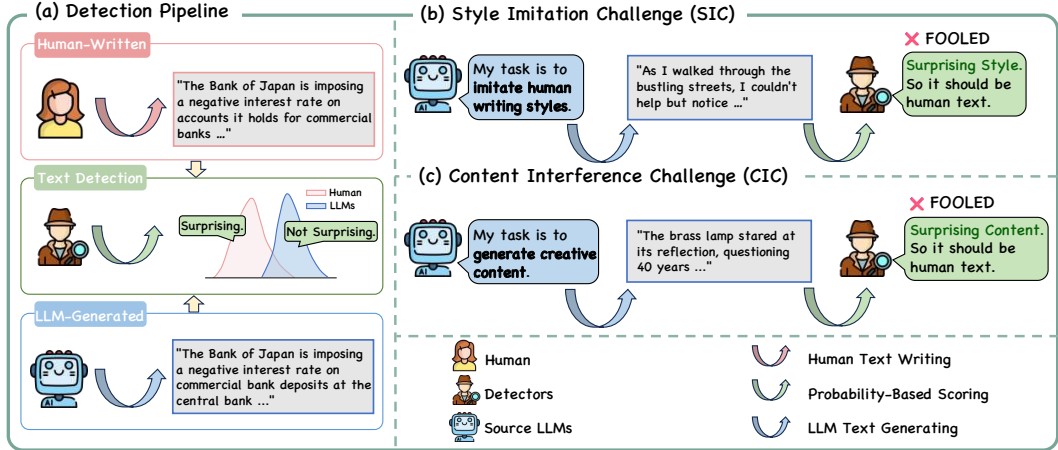

Figure 1: Illustration of challenges in probability-based LLM text detection. (a) Traditional detection pipeline classifies text based on surprise levels to scoring models. (b) Style Imitation Challenge (SIC): LLMs generate text that exhibits surprising stylistic patterns similar to human writing, fooling detectors through style surprise. (c) Content Interference Challenge (CIC): LLMs generate creative content that appears inherently surprising due to content characteristics, misleading detectors to classify it as human-written based on content surprise.

score distributions that enable classification, as illustrated in Figure 1(a). Consequently, these methods demonstrate superior adaptability through next-token prediction-based probability analysis that captures intrinsic linguistic patterns transcending domain boundaries. However, while instructions have demonstrated effectiveness in influencing the characteristics of LLM-generated text (Yin et al., 2024), existing probability-based zero-shot detection approaches predominantly operate on raw text, with only limited exploration of instruction-guided analysis (Bao et al., 2025) that typically employs instructions with minimal relevance to actual textual features.

As LLMs' text generation capabilities continue to advance and gain widespread adoption, we observe and formalize two significant challenges that make successful detection of LLM-generated text difficult: the *style imitation challenge (SIC)* and the *content interference challenge (CIC)*. The SIC arises as LLMs develop increasingly stronger abilities to imitate human writing styles (Chen et al., 2025). As shown in Figure 1(b), the imitative texts mimic human writing styles and poduce probability distributions nearly identical to authentic human-written text, exhibiting similar levels of surprising text to scoring models and successfully evading probability-based detection. The CIC occurs when certain text samples appear inherently surprising to scoring models due to their content characteristics (Hans et al., 2024). Specifically, as shown in Figure 1(c), while human texts naturally exhibit more surprising content, LLMs can also be asked to generate similarly unexpected content through creative or specialized prompts like "Write about a lamp going through a midlife crisis". Under such conditions, the content interference that deviates from world knowledge expectations of scoring models disrupts the detectors to misclassify LLM-generated samples as human-written.

To address these challenges, we propose **SaFT (S**potting **S**tyle Imitation **a**nd **F**iltering Content Interference for Zero-Shot LLM-Generated **T**ext Detection) that addresses style imitation and content interference in probability-based zero-shot detection. SaFT employs instruction-guided probability analysis by introducing *Style-Oriented Instruction Prefix (SOIP)* that explicitly describes text writing styles to guide scoring models in spotting style imitation while filtering content interference. We design two detection components in our framework: *SIC-Detection (SIC-D)* and *CIC-Detection (CIC-D)*. For SIC scenarios where style-imitating LLM texts achieve human-like style surprise, SIC-D prepends candidate text with instruction prefix describing typical human writing style to make them less unexpected. For CIC scenarios where content-related surprise misleads detection, we leverage an intuitive principle: texts can be decomposed into outer style expression and inner content components. Based on this insight, CIC-D computes distributional differences between probability distributions obtained when applying human and LLM style instruction prefixes to the same language model. Since the same language model produces nearly equivalent content surprise

under both instructions, this approach filters out content-induced confounds and preserves style surprise differences for detection. Finally, we integrate SIC-D and CIC-D to compute our SaFT score for robust LLM-generated text detection in realistic deployment scenarios.

To evaluate the practical effectiveness of our approach, we conduct comprehensive experiments under the black-box detection setting, which represents a more challenging yet realistic deployment scenario compared to white-box approaches that require access to the generating model's internal parameters. The results demonstrate the enhanced robustness of SaFT framework under realistic deployment scenarios.

Our main contributions can be summarized as follows:

**1)** We present SaFT, a novel framework built upon Style-Oriented Instruction Prefix (SOIP) that addresses the style imitation challenge and content interference challenge by leveraging style-oriented instructions to guide probability analysis for enhanced LLM-generated text detection.

**2)** We propose SIC-D that spots style imitation by making style-imitating texts less unexpected through probability analysis conditioned on human-style instructions, and CIC-D that filters content interference by difference analysis between probability distributions conditioned on contrasting style instructions, exploiting equivalent content surprises from identical models.

**3)** Extensive experiments on datasets generated by six advanced LLMs across four distinct text domains demonstrate that our method achieves state-of-the-art detection performance compared to existing methods.

## 2 RELATED WORK

Current research in LLM-generated text detection predominantly falls into two paradigms: embedding-based methods and probability-based methods. **Embedding-based methods** typically operate in a supervised manner, involving training binary classifiers on labeled datasets containing both LLM-generated and human-written text samples. These classifiers learn discriminative representations using bag-of-words features (Solaiman et al., 2019) or neural embeddings from models like RoBERTa (Wang et al., 2023; Guo et al., 2024), distinguishing patterns through gradient-based optimization on training data. However, these supervised classifiers often exhibit overfitting tendencies, adapting too closely to the specific distribution of text domains and source models during training, which consequently leads to limited generalization capabilities when exposed to out-of-distribution data (Ghosal et al., 2023; Wu et al., 2025). To address this challenge, our research focuses on probability-based detection, aiming to identify universal features that can be applied across different domains and source models.

**Probability-based methods** primarily operate in a zero-shot manner, relying on statistical features extracted using pre-trained language models as scoring models to extract statistical metrics without requiring training data. Early approaches utilize likelihood scores (Solaiman et al., 2019), entropy (Gehrmann et al., 2019), and log-rank analysis (Su et al., 2023). Recent advances include probability curvature methods like DetectGPT (Mitchell et al., 2023) and its optimized variant Fast-DetectGPT (Bao et al., 2024) with approximately 340 times speedup, and divergence-based approaches like DNA-GPT (Yang et al., 2023). More sophisticated methods have emerged, including Fast-Lastde (Xu et al., 2025), which treats token probability sequences as time series data and employs diversity entropy to quantify temporal dynamics; Binoculars (Hans et al., 2024), which employs a dual-model architecture computing ratios between perplexity and cross-perplexity scores to exploit inter-model consistency patterns; and MOSAIC (Dubois et al., 2025), which combines multiple language models using information-theoretic principles and Blahut-Arimoto optimal weights to achieve robust generator-agnostic detection. While instructions have demonstrated effectiveness in influencing LLM behavior across various domains (Yin et al., 2024), existing probability-based zero-shot detection approaches have only explored instruction-guided detection to a limited extent (Bao et al., 2025). These studies typically employ instructions with minimal relevance to actual textual stylistic features, focusing on generic prompts rather than addressing the fundamental challenges in distinguishing human and LLM-generated texts.

**Differing from** existing approaches that focus on designing statistical metrics from probability distributions, we propose a novel framework utilizing Style-Oriented Instruction Prefix to spot style im-

itation and filter content interference. Our approach is built on the principle that guiding probability analysis toward stylistic recognition while mitigating content influence will substantially improve detection performance for challenging LLM texts.

## 3  METHOD

In this section, we first formulate the problem of probability-based zero-shot LLM-generated text detection in black-box settings. We then introduce our SaFT framework, which systematically addresses the Style Imitation Challenge (SIC) and Content Interference Challenge (CIC) through three core components: Style-Oriented Instruction Prefix (SOIP), SIC-Detection (SIC-D), and CIC-Detection (CIC-D), integrated via a ratio-based formulation to achieve robust detection performance.

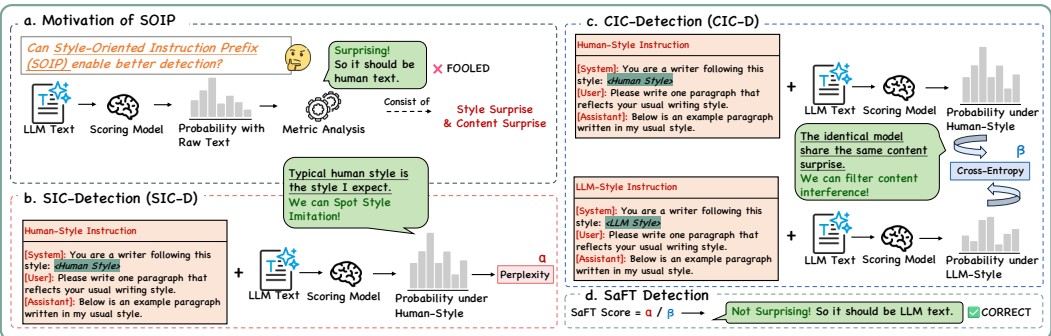

Figure 2: Figure 2: Overview of SaFT framework. (a) Motivation of SOIP: Existing probability-based methods operating solely on raw text struggle with SIC and CIC, where surprise of style and content may fool detectors. (b) SIC-Detection (SIC-D): Uses human-style instruction prefix to detect style imitation, generating perplexity-based $\alpha$ score. (c) CIC-Detection (CIC-D): Applies contrasting human-style and LLM-style instructions to filter content interference, exploiting that identical models share the same content surprise while preserving style surprise differences ($\beta$ score via cross-entropy). (d) SaFT Detection: Combines both scores through SaFT Score = $\alpha$ / $\beta$ for robust classification addressing both challenges.

### 3.1  PROBLEM FORMULATION

We formulate LLM-generated text detection as a binary classification task in the black-box setting. Given a candidate text $t$, our objective is to determine whether it was human-written or LLM-generated without access to the source model.

In the zero-shot setting, detection methods typically design a metric function that maps input text to a real-valued score, leveraging pre-trained language models to extract statistical features without requiring training data. The core assumption underlying these approaches is that human-written texts and LLM-generated texts follow different distributions over the computed metric values. Specifically, human texts often exhibit higher surprise levels (lower probability scores) to scoring models compared to LLM-generated texts, which tend to produce more predictable patterns. Classification is performed by comparing the computed score against a decision threshold $\tau$, where the predicted label $\hat{y}$ indicates whether the text is classified as human-written or LLM-generated. The threshold $\tau$ is typically determined empirically to optimize detection performance across different domains and source models.

### 3.2  SAFT FRAMEWORK

**Style-Oriented Instruction Prefix (SOIP)**. Motivated by the observation that instructions can significantly influence the probabilistic statistical characteristics of LLM outputs Yin et al. (2024), despite limited utilization in existing probability-based detection approaches as illlustrated in Figure 2(a), we pose a key question targeting the SIC and CIC: *can style-oriented instruction prefix*

*enable better detection?* To this end, We employ a chat-based language model $M_\theta$ as our scoring model with instruction prefixes to condition probability computation and enhance feature extraction capabilities. With advancing LLM capabilities in text generation, we observe that current detection methods face two critical challenges: the style imitation challenge (SIC) where LLMs increasingly mimic human writing styles, and the content interference challenge (CIC) where surprising content characteristics interfere with probability analysis. To address these challenges, we propose style-oriented instructions that explicitly describe writing styles as prefixes, and observe that it significantly enhances detection performance by leveraging the model's inherent sensitivity to style-oriented conditioning (See Section 4.3).

**SIC-Detection (SIC-D).** For SIC, we propose SIC-Detection (SIC-D) that targets texts that mimic human writing through style imitation by conditioning evaluation on explicit style instructions, as inllustrated in Figure 2(b). The core idea is spotting style imitation: if a text was generated following human-style guidance, it should exhibit low perplexity under human style instruction $I_h^{\text{SIC}}$. The SIC-D metric is computed as:

$$\alpha(t) = \exp\left(-\frac{1}{n}\sum_{i=1}^{n}\log p_\theta(t_i|I_h^{\text{SIC}}, t_{<i})\right) \tag{1}$$

where $t_i$ is the $i$-th token of $t$ and $n$ is the total number of tokens. This conditioning makes human-style LLM texts more predictable by aligning them with expected human patterns, creating a measurable signature of style imitation that enables detection of texts that would otherwise appear genuinely human. In contrast, regular LLM-generated texts and genuine human texts are less affected by this conditioning. Additionally, while LLM texts trained with human style instructions follow predictable patterns aligned with those instructions, genuine human texts retain their inherent complexity and unpredictability that simple style prompts cannot fully capture, thus remaining surprising to the model.

**CIC-Detection (CIC-D).** For CIC, as shown in Figure 2(c), we propose CIC-Detection (CIC-D) that filters content interference by computing cross-entropy as a measure of distributional differences between probability distributions obtained when applying human and LLM style instruction prefixes to the identical language model. The core insight is that identical models possess the same underlying real-world knowledge, and when applying different style-oriented instructions to the same text, they exhibit nearly equivalent content-related surprise. The cross-entropy computation thus filters content interference, leaving behind "style surprise differentials" that reveal the stylistic patterns. To maximize stylistic contrasts under different instruction prefixes within our detection scope, we define two probability distributions at each position $i$: $p_\theta(\cdot|I_h^{\text{CIC}}, t_{<i})$ conditioned on human-style instruction $I_h^{\text{CIC}}$ and $p_\theta(\cdot|I_m^{\text{CIC}}, t_{<i})$ conditioned on LLM-style instruction $I_m^{\text{CIC}}$.

To focus on high-confidence regions, we apply top-$P$ truncation. Specifically, for each position $i$, we sort all tokens in the vocabulary $\mathcal{V}$ by their probabilities under $p_\theta(\cdot|I_h^{\text{CIC}}, t_{<i})$ in descending order to get $v_1, v_2, \ldots, v_{|\mathcal{V}|}$, where $|\mathcal{V}|$ denotes the vocabulary size. The top-$P$ token set is defined as:

$$\mathcal{V}_P^{(i)} = \left\{ v_k : \sum_{j=1}^{k} p_\theta(v_j|I_h^{\text{CIC}}, t_{<i}) \leq P \text{ or } k = 1 \right\} \tag{2}$$

Both probability distributions are then restricted by setting probabilities to zero for all tokens outside $\mathcal{V}_P^{(i)}$ and renormalizing the remaining probabilities to sum to unity, obtaining the truncated distributions $\hat{p}_\theta(\cdot|I_h^{\text{CIC}}, t_{<i})$ and $\hat{p}_\theta(\cdot|I_m^{\text{CIC}}, t_{<i})$. The CIC-D metric is computed as:

$$\beta(t) = -\frac{1}{n}\sum_{i=1}^{n} \hat{p}_\theta(t_i|I_h^{\text{CIC}}, t_{<i}) \log \hat{p}_\theta(t_i|I_m^{\text{CIC}}, t_{<i}) \tag{3}$$

This truncation restricts both distributions to the same high-confidence token set determined by the human-style distribution, emphasizing regions where style surprise differences are most pronounced while reducing noise from low-probability tokens. Since both distributions evaluate identical content using the same scoring model, content-related surprise factors affect both equivalently, allowing the cross-entropy to extract style surprise differences while filtering content interference effects.

**Score Integration and Decision Rule.** Since CIC-D captures the differential between style-related surprises under contrasting instructions rather than style surprise itself, such differentials alone are

difficult to serve as effective classification features, as verified by corresponding experiments in Section 4.3. Therefore, as shown in Figure 2(d), we integrate the components through an empirically-determined ratio formulation that leverages CIC-D as a modulating factor to guide SIC-D: this enables the overall score to filter content interference while simultaneously spotting style imitation:

$$\text{SaFT}(t) = \frac{\alpha(t)}{\beta(t)} \tag{4}$$

where $\alpha(t)$ represents SIC-D and $\beta(t)$ represents CIC-D. The detection decision follows:

$$\hat{y} = \begin{cases} \text{LLM-generated} & \text{if } \text{SaFT}(t) > \tau \\ \text{Human-written} & \text{otherwise} \end{cases} \tag{5}$$

## 4 EXPERIMENTS

### 4.1 SETTINGS

**Task Definition.** Given a text sample $t$, our task is to determine whether it was written by humans or generated by LLMs, outputting a binary classification decision. We treat LLM-generated texts as positive samples and human-written texts as negative samples. We operate under black-box detection constraints where detectors cannot access the generating model's internal parameters or probability distributions, reflecting realistic deployment scenarios where generated text may originate from proprietary or unknown systems. Detailed black-box setting specifications are provided in Appendix A.1.

**Source Models.** The source models we selected represent the latest and most advanced LLMs from leading LLM companies. Our selection includes the two most advanced versions from each major provider: *Claude-4 Sonnet* (claude-sonnet-4-20250514) and *Claude-4 Opus* (claude-opus-4-20250514) (Anthropic, 2025) from Anthropic, *Gemini-2.5 Flash* (gemini-2.5-flash) and *Gemini-2.5 Pro* (gemini-2.5-pro) (Comanici et al., 2025) from Google, as well as *GPT-4o* (gpt-4o) (OpenAI, 2024) and *GPT-4.1* (gpt-4.1) [2] from OpenAI. For dataset preparation, we employ the ChatCompletion API [3] across all selected models.

**Datasets.** We conduct comprehensive evaluation across four diverse domains that represent high-risk scenarios for LLM misuse, including *XSum* (Narayan et al., 2018) for news content, *ArXiv Abstracts* [4] for academic writing, *PubMedQA* (Jin et al., 2019) for biomedical research, and *Yelp Reviews* [5] for informal consumer content. These datasets provide varied linguistic styles and domain-specific characteristics essential for robust detection evaluation. For each dataset, we ensure balanced evaluation by randomly selecting 150 human-written samples and generating equal numbers of corresponding LLM texts using the same prefix (30 tokens for articles or questions for PubMedQA). Detailed dataset descriptions and generation procedures are provided in Appendix B.1 and Appendix B.2 respectively.

**Baselines.** We compare our approach against 11 representative detection methods spanning both traditional statistical approaches and recent state-of-the-art techniques. These include fundamental statistical methods such as *Likelihood* (Solaiman et al., 2019), *Entropy* (Gehrmann et al., 2019; Ippolito et al., 2020), *LogRank* (Solaiman et al., 2019), and *LRR*; perturbation-based approaches including *NPR* (Su et al., 2023), *DetectGPT* (Mitchell et al., 2023), and *Fast-DetectGPT* (Bao et al., 2024); advanced methods like *DNA-GPT* (Yang et al., 2023), *Binoculars* (Hans et al., 2024), *Fast-Lastde* (Xu et al., 2025), and the recent *MOSAIC* (Dubois et al., 2025) ensemble method. Our baseline selection covers sample-based approaches that analyze individual probability distributions and distribution-based methods that examine statistical patterns across text samples. Complete baseline descriptions and implementation details are provided in Appendix A.3 and Appendix A.4 respectively.

**Evaluation Metrics.** Following standard practice in detection research, we adopt AUROC to assess overall discriminative performance across all decision thresholds, providing threshold-independent

---

[2]https://openai.com/index/gpt-4-1

[3]https://platform.openai.com/docs/guides/text-generation/chat-completions-api

[4]https://www.kaggle.com/datasets/spsayakpaul/arxiv-paper-abstracts/data

[5]https://www.yelp.com/dataset_challenge

evaluation of separation capability between human-written and LLM-generated text. Additionally, we report TPR @ 10% FPR to evaluate performance at practically relevant operating points where controlling false positives on human text is crucial for real-world deployment scenarios. Detailed metric definitions are available in Appendix A.2.

**Hyperparameter Configuration.** For our SaFT framework, we employ top-$P$ truncation with $P = 0.4$ for cross-entropy computation in the CIC-D component, which focuses analysis on high-confidence token predictions while filtering low-probability noise. The style instructions employed in our framework ($I_h^{\text{SIC}}$, $I_h^{\text{CIC}}$, and $I_m^{\text{CIC}}$) are specifically designed to capture distinctive writing characteristics, with complete instruction templates provided in Appendix C. These hyperparameter choices are validated through the ablation studies presented in Section 4.3.

## 4.2 MAIN RESULTS

**Overall Performance.** Table 1 presents the comprehensive performance comparison across six advanced LLMs, where our SaFT framework consistently achieves the best performance in both AUROC and TPR @ 10% FPR across all evaluated models. SaFT demonstrates substantial improvements over the second-best baselines, with AUROC gains ranging from 2.8% to 9.9% (average 4.9%) and more pronounced TPR improvements ranging from 11.2% to 45.3% (average 20.4%). For all 12 model-metric combinations, SaFT provides the most accurate detection performance, outperforming strong baselines such as MOSAIC and Binoculars. Notably, the TPR improvements are more pronounced than AUROC gains, indicating that our instruction design particularly enhances the framework's ability to maintain high true positive rates while controlling false positives, which is crucial for practical deployment scenarios. Furthermore, while previous methods show significant performance variations across different models, with some methods like MOSAIC performing excellently on Claude models but less effectively on certain Gemini variants, SaFT maintains robust detection performance regardless of the source model, validating the generalizability of our style-oriented instruction approach.

**Inference Efficiency.** SaFT achieves competitive efficiency at 0.34 s/1k words while delivering state-of-the-art detection performance. This efficiency enables practical deployment while maintaining superior accuracy across all evaluated models and datasets. Detailed efficiency analysis are provided in Appendix D with specific inference speeds in Table 4.

Table 1: Detection results for text generated by *Claude-4-Sonnet*, *Claude-4-Opus*, *Gemini-2.5-Flash*, *Gemini-2.5-Pro*, *GPT-4o*, and *GPT-4.1* models, which are averaged across *XSum*, *ArXiv*, *PubMed*, and *Yelp* datasets. The metrics are *AUROC* and *TPR* calculated at 10% FPR. The best and second-best results in each column are marked with **bold** and underline respectively. The "*(Imp↑)*" row indicates the score improvement upon the second-best baselines. More detailed detection results are available in Table 5, Table 6, and Table 7 in Appendix G.

| Method | Claude-4-Sonnet | | Claude-4-Opus | | Gemini-2.5-Flash | | Gemini-2.5-Pro | | GPT-4o | | GPT-4.1 | |
| --- | --- | --- | --- | --- | --- | --- | --- | --- | --- | --- | --- | --- |
| | *AUROC* | *TPR* | *AUROC* | *TPR* | *AUROC* | *TPR* | *AUROC* | *TPR* | *AUROC* | *TPR* | *AUROC* | *TPR* |
| Likelihood | 0.6802 | 0.3033 | 0.6777 | 0.3033 | 0.7588 | 0.4450 | 0.8179 | 0.5467 | 0.8456 | 0.6417 | 0.8314 | 0.5900 |
| Entropy | 0.4190 | 0.0750 | 0.4212 | 0.0700 | 0.2864 | 0.0317 | 0.2616 | 0.0267 | 0.2283 | 0.0317 | 0.2245 | 0.0200 |
| LogRank | 0.6678 | 0.2750 | 0.6623 | 0.2717 | 0.7248 | 0.3617 | 0.7937 | 0.4583 | 0.8215 | 0.5650 | 0.7986 | 0.4717 |
| LRR | 0.6127 | 0.1817 | 0.5989 | 0.1617 | 0.5631 | 0.1483 | 0.6423 | 0.2600 | 0.6518 | 0.2317 | 0.6222 | 0.1933 |
| NPR | 0.6664 | 0.3350 | 0.6619 | 0.3333 | 0.5461 | 0.1783 | 0.4829 | 0.1367 | 0.5426 | 0.2067 | 0.5517 | 0.2150 |
| DNA-GPT | 0.6575 | 0.2550 | 0.6556 | 0.2367 | 0.6266 | 0.1983 | 0.6577 | 0.2033 | 0.7388 | 0.3800 | 0.7109 | 0.2817 |
| DetectGPT | 0.7279 | 0.3650 | 0.7117 | 0.3683 | 0.4307 | 0.0583 | 0.3629 | 0.0367 | 0.4656 | 0.0833 | 0.4562 | 0.1333 |
| Fast-DetectGPT | 0.7030 | 0.3433 | 0.6970 | 0.3417 | 0.6828 | 0.3167 | 0.7902 | 0.4267 | 0.8032 | 0.5000 | 0.7405 | 0.3450 |
| Fast-Lastde | 0.7991 | 0.5183 | 0.8034 | 0.5217 | 0.7168 | 0.3767 | 0.7600 | 0.4367 | 0.7693 | 0.4667 | 0.7825 | 0.4550 |
| Binoculars | 0.8838 | 0.6667 | 0.8833 | 0.6567 | 0.8280 | 0.5367 | 0.8932 | 0.7108 | 0.9343 | 0.8133 | 0.8995 | 0.7050 |
| MOSAIC | 0.9331 | 0.8189 | 0.9338 | 0.8231 | 0.8628 | 0.6012 | 0.8898 | 0.6803 | 0.9401 | 0.8244 | 0.9208 | 0.7549 |
| **SaFT (Ours)** | **0.9672** | **0.9117** | **0.9650** | **0.9150** | **0.9481** | **0.8733** | **0.9183** | **0.8000** | **0.9703** | **0.9517** | **0.9795** | **0.9567** |
| *(Imp↑)* | *3.7%* | *11.3%* | *3.3%* | *11.2%* | *9.9%* | *45.3%* | *2.8%* | *12.5%* | *3.2%* | *15.4%* | *6.4%* | *26.7%* |

## 4.3 ABLATION STUDY

**Instruction Component Ablation.** To validate the effectiveness and specificity of our SOIP approach, we examine SaFT performance across different instruction component configurations using $I_h^{\text{SIC}}$, $I_h^{\text{CIC}}$, and $I_m^{\text{CIC}}$. Note that configurations where both $I_h^{\text{CIC}}$ and $I_m^{\text{CIC}}$ are absent would result in

identical distributions for CIC-D, leading to zero cross-entropy and division by zero in the final ratio, hence such cases are not evaluated. Table 2 reveals two critical insights about our framework design. First, $I_h^{\text{SIC}}$ is indispensable for effective detection: without it (rows 1-3), performance remains limited with average AUROC scores ranging from 0.8937 to 0.9357 and TPR scores averaging only 0.6700 to 0.8267. This validates our theoretical foundation that SIC-D requires human-style conditioning to identify style imitation patterns. Second, CIC-D requires contrasting style instructions for optimal content filtering: comparing row 4 ($I_h^{\text{SIC}} + I_m^{\text{CIC}}$) and row 5 ($I_h^{\text{SIC}} + I_h^{\text{CIC}}$), both achieve similar AUROC performance (0.9608 vs 0.9521) but show different TPR patterns (0.8944 vs 0.8633), while the full configuration (row 6) reaches optimal performance with 0.9714 average AUROC and exceptional 0.9622 average TPR. These systematic performance degradations across different ablation configurations demonstrate that our SOIP design is principled rather than arbitrary, with each instruction component serving a specific and necessary role in the overall detection framework.

Table 2: Instruction component ablation results for SaFT on text generated by six advanced LLMs on XSum dataset. We compare different instruction component configurations using $I_h^{\text{SIC}}$, $I_h^{\text{CIC}}$, and $I_m^{\text{CIC}}$. The best results in each column are marked with **bold**.

| Variant | | | Claude-4-Sonnet | | Claude-4-Opus | | Gemini-2.5-Flash | | Gemini-2.5-Pro | | GPT-4o | | GPT-4.1 | |
|---|---|---|---|---|---|---|---|---|---|---|---|---|---|---|
| $I_h^{\text{SIC}}$ | $I_h^{\text{CIC}}$ | $I_m^{\text{CIC}}$ | AUROC | TPR | AUROC | TPR | AUROC | TPR | AUROC | TPR | AUROC | TPR | AUROC | TPR |
| × | × | ✓ | 0.9118 | 0.7533 | 0.9037 | 0.7533 | 0.8833 | 0.6400 | 0.8801 | 0.6333 | 0.9612 | 0.9333 | 0.9453 | 0.8600 |
| × | ✓ | × | 0.8793 | 0.6333 | 0.8700 | 0.6400 | 0.8896 | 0.6467 | 0.8491 | 0.5000 | 0.9271 | 0.7467 | 0.9470 | 0.8533 |
| × | ✓ | ✓ | 0.9388 | 0.8200 | 0.9252 | 0.7867 | 0.9290 | 0.7867 | 0.8961 | 0.7133 | 0.9676 | 0.9600 | 0.9572 | 0.8933 |
| ✓ | × | ✓ | 0.9733 | 0.9267 | 0.9639 | 0.8733 | 0.9466 | 0.8267 | 0.9202 | 0.7867 | 0.9809 | 0.9800 | 0.9800 | 0.9733 |
| ✓ | ✓ | × | 0.9628 | 0.8933 | 0.9527 | 0.8733 | 0.9521 | 0.8667 | 0.9032 | 0.7067 | 0.9618 | 0.8733 | 0.9799 | 0.9667 |
| ✓ | ✓ | ✓ | **0.9826** | **1.0000** | **0.9762** | **0.9733** | **0.9715** | **0.9800** | **0.9333** | **0.8333** | **0.9823** | **1.0000** | **0.9828** | **0.9867** |

**Score Component Ablation.** SaFT score consists of two components: SIC-D ($\alpha(t)$) and CIC-D ($\beta(t)$). To demonstrate the indispensable role of both components in SaFT, we conducted ablation experiments on XSum texts generated by the six advanced LLMs. We compare three configurations: (1) Only SIC-D using $\alpha(t)$ alone, (2) Only CIC-D using $\beta(t)$ alone, and (3) SaFT (Full) using the ratio $\alpha(t)/\beta(t)$. Table 3 reveals that relying solely on SIC-D or CIC-D can only yield limited detection performance compared to their integration. SIC-D alone already demonstrates strong detection capabilities with an average AUROC of 0.9511 across all models, significantly outperforming most existing baselines, yet its TPR performance varies significantly across models (0.7367-0.9733). CIC-D alone shows substantially weaker performance, with an average AUROC of only 0.8119 and particularly poor TPR performance (0.2600-0.7533), confirming our hypothesis that style surprise differentials alone are insufficient as classification features. In stark contrast, the full SaFT framework achieves exceptional detection performance with an average AUROC of 0.9715, representing a 2.1% improvement over SIC-D alone and a remarkable 19.6% improvement over CIC-D alone. Most notably, the integrated approach achieves perfect TPR (1.0000) for Claude-4-Sonnet and GPT-4o while maintaining AUROC scores above 0.98, demonstrating that while individual components are insufficient for optimal detection, their synergistic combination produces exceptional detection capabilities that neither component can achieve alone.

Table 3: Component ablation results for SaFT on text generated by six advanced LLMs on XSum dataset. We compare the full SaFT method with its individual components. The best results in each column are marked with **bold**.

| Variant | | Claude-4-Sonnet | | Claude-4-Opus | | Gemini-2.5-Flash | | Gemini-2.5-Pro | | GPT-4o | | GPT-4.1 | |
|---|---|---|---|---|---|---|---|---|---|---|---|---|---|
| SIC-D | CIC-D | AUROC | TPR | AUROC | TPR | AUROC | TPR | AUROC | TPR | AUROC | TPR | AUROC | TPR |
| ✓ | × | 0.9562 | 0.8567 | 0.9501 | 0.8633 | 0.9171 | 0.7367 | 0.9227 | 0.7644 | **0.9834** | 0.9733 | 0.9771 | 0.9467 |
| × | ✓ | 0.7595 | 0.3267 | 0.7734 | 0.3533 | 0.6907 | 0.2600 | 0.8433 | 0.6000 | 0.9195 | 0.7533 | 0.8852 | 0.6533 |
| ✓ | ✓ | **0.9826** | **1.0000** | **0.9762** | **0.9733** | **0.9715** | **0.9800** | **0.9333** | **0.8333** | 0.9823 | **1.0000** | **0.9828** | **0.9867** |

**Top-$P$ Truncation Ablation.** We evaluate the impact of top-$P$ parameter across $P$ values of $\{0, 0.2, 0.4, 0.6, 0.8, 1.0\}$ using GPT-4.1 as the source model across four datasets, as shown in Figure 3. The results reveal distinct patterns across domains: while ArXiv and Yelp maintain stable performance across all $P$ values, PubMed and XSum exhibit degradation when $P = 1.0$. This domain-specific sensitivity demonstrates that top-$P$ truncation effectively filters noise from low-probability tokens

that can interfere with style signal extraction, particularly in specialized domains. The choice of $P = 0.4$ represents an optimal balance, residing in the stable performance region across all four datasets while avoiding potential degradation from including the full vocabulary. These findings validate that focusing cross-entropy computation on high-confidence tokens enhances the reliability of style surprise difference detection across diverse text domains.

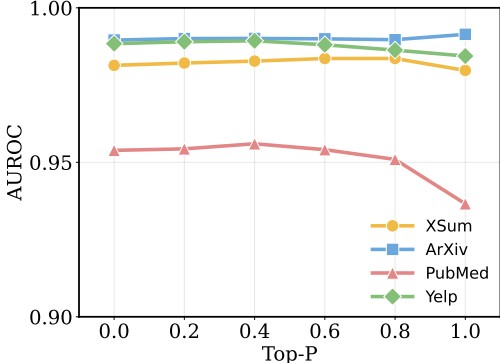
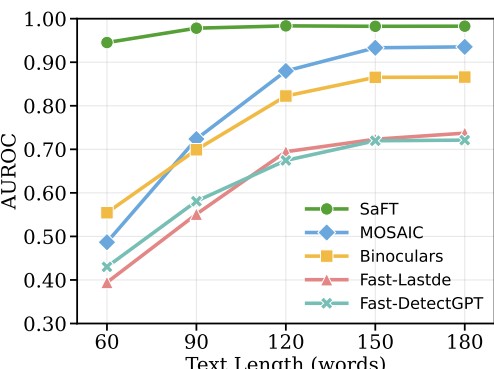

Figure 3: Top-$P$ ablation study for SaFT framework on four datasets using GPT-4.1 as the source model, showing AUROC performance under varying $P$ values.

Figure 4: Detection performance across varying text lengths on XSum (GPT-4.1) dataset, comparing SaFT with four state-of-the-art methods. The metric is AUROC.

### 4.4 ROBUSTNESS ANALYSIS AND DISCUSSION

**Text Length Robustness.** Previous research (Chakraborty et al., 2024; Wu et al., 2024) has established that detection accuracy typically decreases with shorter text samples due to limited statistical evidence. We assess SaFT's robustness across different text lengths by evaluating performance on XSum texts generated by GPT-4.1, truncated to {60, 90, 120, 150, 180} words. As shown in Figure 4, SaFT consistently outperforms all baselines across all text lengths, with the performance gap most pronounced at shorter lengths where competing methods exhibit steep upward curves indicating strong length dependency. SaFT demonstrates exceptional robustness to text length variations, achieving near-optimal performance even at 60 words.

**Paraphrasing Attack Robustness.** Following previous work (Bao et al., 2024; Xu et al., 2025), we evaluate robustness against paraphrasing attacks using T5-Paraphraser to test whether our method can resist adversarial modifications that preserve semantic content while altering surface expressions. As shown in Figure 5, SaFT demonstrates superior robustness with minimal performance degradation compared to baseline methods that experience substantial drops. See Appendix E for detailed comparison and analysis.

**Limitations.** While our instruction design discussed in Appendix C is grounded in empirical findings about style surprise differences, it focuses primarily on a single contrast dimension and may not capture other effective distinctions. Future work could systematically explore multi-dimensional instruction designs and automated optimization techniques to enhance detection performance.

## 5 CONCLUSION

We present SaFT, a novel detection framework built upon Style-Oriented Instruction Prefix (SOIP) that addresses the style imitation challenge (SIC) and content interference challenge (CIC). Our approach proposes SIC-Detection (SIC-D) for spotting style imitation via human-style instruction conditioning and CIC-Detection (CIC-D) for filtering content interference through probability distributional analysis between contrasting style instructions. Their ratio-based combination produces robust detection capabilities across diverse scenarios. Results demonstrate that SaFT significantly outperforms existing state-of-the-art probability-based zero-shot methods.

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

# A EXPERIMENTAL SETTINGS

## A.1 BLACK-BOX DETECTION TASK DEFINITION

In this work, we adopt the black-box detection setting to evaluate LLM-generated text detectors. Unlike white-box approaches that require access to the generating model's internal parameters and probability distributions, black-box detection operates under the realistic constraint that detectors cannot access the source model that produced the text. This setting better reflects practical deployment scenarios where generated content may originate from proprietary or unknown systems.

Formally, given a text sample $t$ and a collection of available proxy models, the black-box detection task aims to determine whether $t$ was LLM-generated without knowledge of the specific model or parameters used in its creation. This constraint necessitates the development of detection methods that rely on learned patterns and statistical signatures robust across various generation sources, rather than exploiting model-specific artifacts or direct probability computations, making it a more challenging but practically relevant problem compared to white-box detection.

## A.2 EVALUATION METRICS

**AUROC.** We adopt the Area Under the Receiver Operating Characteristic curve (AUROC) to evaluate the overall discriminative power of our detection method. This threshold-independent metric aggregates performance across the entire spectrum of decision boundaries, providing a comprehensive assessment of how well the model separates human-written and LLM-generated content, with values ranging from 0 to 1. Higher AUROC values indicate superior classification performance, with AUROC = 0.5 representing random guessing and perfect discrimination achieved at AUROC = 1.0.

**TPR @ 10% FPR.** To evaluate performance at a practically relevant operating point, we report the True Positive Rate (TPR) when the False Positive Rate (FPR) is fixed at 10%. This metric captures the model's sensitivity in detecting LLM-generated texts while maintaining an acceptable level of false alarms on human-written content, providing insight into real-world deployment scenarios where controlling false positives is crucial.

## A.3 BASELINE DETECTORS

**Likelihood** (Solaiman et al., 2019). This fundamental approach computes the average log-probability of tokens in the candidate text using a pre-trained language model. The underlying

assumption is that LLM-generated texts exhibit higher likelihood scores compared to human-written content, as they follow more predictable patterns learned during training.

**Entropy** (Gehrmann et al., 2019; Ippolito et al., 2020).This method leverages the information-theoretic concept of entropy to measure textual randomness. It calculates the average entropy across token probability distributions, exploiting the observation that human writing typically demonstrates higher unpredictability and variability than LLM-generated content.

**LogRank** (Solaiman et al., 2019). This approach assigns ranking scores to tokens based on their probability positions within the model's vocabulary distribution. The method computes logarithmic rankings for each token given its preceding context, with higher average scores indicating stronger likelihood of machine generation.

**LRR** (Su et al., 2023). The Log-Likelihood to Log-Rank Ratio represents a composite metric that combines token likelihood and ranking information. This method aims to capture both the probability magnitude and relative positioning of tokens to improve detection sensitivity.

**NPR** (Su et al., 2023). The Normalized Perturbed Log-Rank method introduces controlled perturbations to the input text and analyzes the resulting changes in log-rank scores. This technique exploits the differential sensitivity of human versus LLM-generated texts to minor modifications.

**DNA-GPT** (Yang et al., 2023). This approach utilizes N-gram statistical analysis in combination with probability divergence measures. The method compares original text segments with model-completed versions, leveraging the hypothesis that LLMs exhibit characteristic completion patterns distinct from human writing.

**DetectGPT** (Mitchell et al., 2023). This method employs probabilistic curvature analysis by introducing random perturbations to the candidate text. It discriminates between human and LLM-generated content by examining the curvature properties of log-probability distributions around the original text.

**Fast-DetectGPT** (Bao et al., 2024). An optimized variant of DetectGPT that achieves comparable detection accuracy while significantly reducing computational overhead. This method maintains the core curvature-based detection principle while employing efficient approximation techniques for practical deployment.

**Binoculars** (Hans et al., 2024). This detector employs a dual-model architecture that computes the ratio between perplexity and cross-perplexity scores. The method evaluates how one language model responds to token predictions from another model, exploiting inter-model consistency patterns to identify LLM-generated text.

**Fast-Lastde** (Xu et al., 2025). This detector combines local and global statistical features by treating token probability sequences as time series data. The method employs diversity entropy to quantify temporal dynamics within probability sequences. Fast-Lastde represents an optimized variant designed for real-time detection through efficient sampling strategies while maintaining robust detection performance.

**MOSAIC** (Dubois et al., 2025). This ensemble method combines multiple language models using information-theoretic principles for robust generator-agnostic detection. It employs Blahut-Arimoto optimal weights to aggregate probability distributions from diverse LLMs, providing scalable detection across various domains without requiring validation datasets.

Notably, while Glimpse (Bao et al., 2025) demonstrates promising results in preliminary explorations of instruction-based applications, its dependence on expensive LLM APIs constrains its practical deployment, thus we do not consider it as a baseline detector in our evaluation.

### A.4 DETECTOR SETTINGS

All experiments are conducted on two RTX 3090 GPUs. We configure the baseline detectors using their standard model configurations to ensure fair comparison. The Neo-2.7B (Black et al., 2021) serves as the primary scoring model for the majority of methods, including Likelihood, Entropy, LogRank, LRR, DNA-GPT, NPR, DetectGPT, and Fast-DetectGPT. Among these, perturbation-based approaches require additional models: NPR and DetectGPT incorporate T5-3B (Chen et al., 2019)

for generating text perturbations, while Fast-DetectGPT employs GPT-J (Wang & Komatsuzaki, 2021) as a surrogate model for efficient sampling.

The remaining detectors adopt distinct model architectures. Fast-Lastde operates independently with GPT-J for probability scoring. Binoculars implements a dual-model architecture using the Falcon family (Almazrouei et al., 2023) (Falcon-7B and Falcon-7B-instruct) as observer and performer respectively. MOSAIC adopts an ensemble strategy with Llama family variations (Llama-2-7B, Llama-2-7B-chat, TowerBase-7B and TowerBase-13B (Alves et al., 2024)). Our proposed SaFT method utilizes Llama-3.1-8B-Instruct (Grattafiori et al., 2024) as the scoring model.

All implementations use full-precision (float32) computation, except for Binoculars, MOSAIC, and SaFT which employ half-precision (float16) for computational efficiency.

## B   DATASET CREATION

### B.1   DOMAINS

We evaluate our method on four datasets representing high-risk domains where LLM misuse could have significant consequences. These datasets span academic writing, journalism, medical literature, and consumer reviews, providing diverse linguistic styles and domain-specific characteristics.

**XSum** (Narayan et al., 2018). This dataset contains BBC news articles spanning 2010-2017, encompassing approximately 227K documents across multiple domains including politics, technology, entertainment, and current affairs. The articles represent professional journalistic writing with formal structure and factual content presentation.

**ArXiv Abstracts** [6]. We utilize machine learning paper abstracts from the arXiv repository, covering publications from 2007-2020. This academic corpus contains over 100K research documents characterized by technical vocabulary, formal academic discourse, and specialized scientific terminology.

**PubMedQA** (Jin et al., 2019). This biomedical dataset comprises question-answer pairs extracted from medical literature abstracts. It represents highly specialized scientific writing with domain-specific terminology, clinical language patterns, and evidence-based reasoning structures typical of medical research.

**Yelp Reviews** [7]. Consumer review data from the Yelp platform provides examples of informal, conversational writing styles. The dataset includes approximately 600K reviews with sentiment labels, characterized by colloquial language, personal opinions, and varied writing quality reflecting diverse user backgrounds.

### B.2   GENERATION PROCESS

All text generation tasks were conducted through chat with LLMs using a temperature setting of 0.8 to introduce stylistic variation while maintaining coherence. We designed domain-specific system roles and generation prompts to capture the distinct writing characteristics of each dataset. For news articles (XSum), we instructed the model to adopt a news writer persona; for academic abstracts (ArXiv), an academic writer role was assigned; restaurant reviews (Yelp) utilized a review writer persona; and biomedical content (PubMed) employed a technical writer role. Each generation prompt specified the desired output length and format appropriate to the domain's conventions. Below, we provide the generation instructions for texts in different domains:

---

[6]https://www.kaggle.com/datasets/spsayakpaul/arxiv-paper-abstracts/data
[7]https://www.yelp.com/dataset_challenge

**Message for XSum**

```
[
  {'role': 'system', 'content': 'You are a News writer.'},
  {'role': 'user', 'content': 'Please write an article with about
    150 words starting exactly with: <prefix>'},
]
```

The `<prefix>` could be like "Joshua King needed some assistance from Newcastle's Steven Taylor to create the opening for Bournemouth's first goal", and the response is supposed to start with it.

**Message for ArXiv**

```
[
  {'role': 'system', 'content': 'You are a Academic writer.'},
  {'role': 'user', 'content': 'Please write an article with about
    150 words starting exactly with: <prefix>'},
]
```

The `<prefix>` could be like "In this paper, we introduce a scanner package enhanced by deep learning (DL) techniques", and the response is supposed to start with it.

**Message for Yelp**

```
[
  {'role': 'system', 'content': 'You are a Restaurant Review writer
    .'},
  {'role': 'user', 'content': 'Please write an article with about
    150 words starting exactly with: <prefix>'},
]
```

The `<prefix>` could be like "I guess this particular restaurant benefits from people who work at the Waterfront stopping in after a shift", and the response is supposed to start with it.

**Message for PubMed**

```
[
  {'role': 'system', 'content': 'You are a Technical writer.'},
  {'role': 'user', 'content': 'Please answer the question in about
    50 words. <prefix>'},
]
```

The `<prefix>` could be like "Question: Can communication with terminally ill patients be taught? Answer:" and the response is supposed to answer the question directly.

## C  DISCUSSION ON INSTRUCTION SETTINGS

Recent empirical and computational studies have systematically investigated style surprise differences between human-written and LLM-generated texts through the lens of cognitive load theory. (Sweller, 2011) established that human working memory has severe capacity limitations, necessitating efficient, concise communication to minimize cognitive load. (Grice, 1975) demonstrated that human communication follows cooperative principles including the maxim of brevity, reflecting evolved constraints of human information processing. (Reinhart et al., 2025) found that LLMs systematically violate these cognitive efficiency principles, using participial clauses and nominalizations at substantially higher rates than humans, resulting in informationally dense text that lacks

human conciseness constraints. These studies demonstrate that human writing reflects cognitive load optimization through concise expression, whereas LLM-generated writing favors detailed elaboration unconstrained by working memory limitations.

Building on these findings, we carefully designed the style-oriented instructions used in our approach, as shown in the following conversational templates. Note that $I_h^{\text{SIC}}$ and $I_h^{\text{CIC}}$ use the same human-style instruction that encourages concise expression, while $I_m^{\text{CIC}}$ employs an LLM-style instruction that emphasizes detailed elaboration. The choice of "concise" versus "detailed" instructions directly operationalizes cognitive load theory: human writers, constrained by working memory limitations, favor concise expression, while LLMs, unconstrained by cognitive load, tend toward detailed elaboration. These templates direct the model's probability computation toward detecting style imitation while enabling content interference filtering through our framework's dual-component design.

---

**Human-style Instruction ($I_h^{\text{SIC}}$ & $I_h^{\text{CIC}}$)**

System: You are a writer following this style: Express ideas using concise sentences.
User: Please write one paragraph that reflects your usual writing style.
Assistant: Below is an example paragraph written in my usual style.

---

**LLM-style Instruction ($I_m^{\text{CIC}}$)**

System: You are a writer following this style: Express ideas using detailed sentences.
User: Please write one paragraph that reflects your usual writing style.
Assistant: Below is an example paragraph written in my usual style.

---

## D  EFFICIENCY ANALYSIS

Our SaFT framework demonstrates competitive computational efficiency while achieving superior detection performance. As shown in Table 4, SaFT requires 0.34 s/1k words, positioning it among the most efficient and effective methods in the zero-shot detection paradigm. This efficiency stems from our SOIP approach that requires only three forward passes over the identical language model without complex computational operations.

Table 4: Inference efficiency comparison of zero-shot detection methods.

| Method | Inference Speed (s/1k words) |
|---|---|
| Likelihood | 0.29 |
| Entropy | 0.29 |
| LogRank | 0.30 |
| Fast-DetectGPT | 0.33 |
| **SaFT (Ours)** | **0.34** |
| Fast-Lastde | 0.42 |
| LRR | 0.58 |
| Binoculars | 0.69 |
| MOSAIC | 9.77 |
| DNA-GPT | 19.88 |
| DetectGPT | 33.42 |
| NPR | 34.67 |

The computational landscape of zero-shot detection methods reveals distinct efficiency patterns. Basic statistical methods such as Likelihood (0.29 s/1k words), Entropy (0.29 s/1k words), and LogRank (0.30 s/1k words) achieve the fastest inference times through single forward passes, but at the cost of substantially lower detection accuracy. Traditional perturbation-based approaches demonstrate significantly higher computational overhead (DetectGPT: 33.42 s/1k words, NPR: 34.67 s/1k words), while Fast-DetectGPT achieves substantial speedup at 0.33 s/1k words through efficient sampling strategies. DNA-GPT requires 19.88 s/1k words for its divergence-based analysis. Among

sophisticated methods, MOSAIC exhibits high computational cost at 9.77 s/1k words, reflecting its ensemble approach. Binoculars requires 0.69 s/1k words and Fast-Lastde requires 0.42 s/1k words for their respective dual-model and time-series approaches.

Compared to the second-best baseline MOSAIC, SaFT achieves a 28.7× speedup while delivering consistent accuracy improvements across all evaluated models and datasets. This efficiency-accuracy balance makes SaFT particularly suitable for real-world deployment scenarios where both detection quality and computational constraints are critical considerations.

# E PARAPHRASING ATTACK ROBUSTNESS ANALYSIS

**Experimental Setup.** Following established protocols (Bao et al., 2024; Xu et al., 2025), we evaluate robustness against paraphrasing attacks using the T5-Paraphraser model to transform LLM-generated texts while preserving semantic content. The T5-Paraphraser applies sentence-level paraphrasing to alter surface expressions and syntactic structures. Additionally, we introduce controlled disruptions by randomly swapping adjacent word pairs in sentences exceeding 20 words, mimicking realistic adversarial modifications. We conduct experiments on ArXiv (GPT-4.1), comparing detection performance on both original and paraphrased versions, where the combined operations simulate comprehensive surface-level attacks.

**Results and Analysis.** Figure 5 demonstrates that SaFT exhibits exceptional robustness with only a 3.5% AUROC degradation, substantially outperforming baseline methods that experience 6.2%-13.7% drops. SaFT's SOIP approach focuses on stylistic characteristics that appear less affected by surface-level paraphrasing modifications compared to traditional methods that operate without instruction prefixes. This difference in detection mechanisms contributes to SaFT's maintained performance when texts undergo lexical and syntactic transformations.

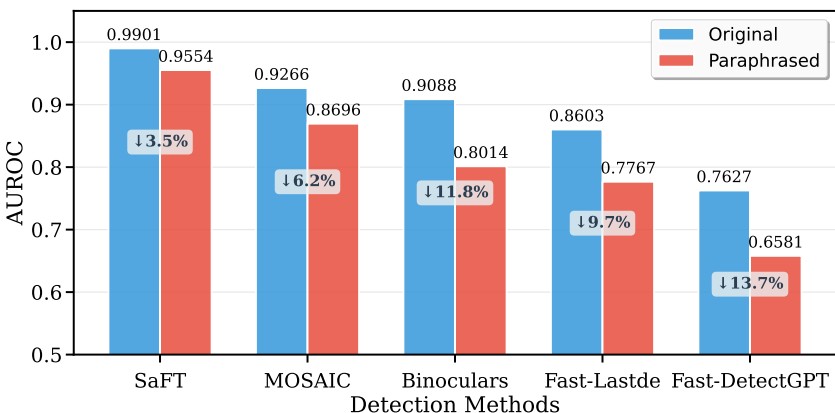

Figure 5: Detection performance on original and paraphrased texts using ArXiv (GPT-4.1) dataset. SaFT demonstrates superior robustness against paraphrasing attacks compared to baseline methods.

# F USE OF LARGE LANGUAGE MODELS

Large Language Models were used for language polishing and grammatical refinement of the manuscript.

# G FULL RESULTS

Table 5: Detailed detection results for text generated by *Claude-4-Sonnet* and *Claude-4-Opus* models. Results are reported as "*AUROC / TPR*", where TPR is calculated at 10% FPR. The best results in each column are marked with **bold**.

| Method | Claude-4-Sonnet | | | | Claude-4-Opus | | | |
| --- | --- | --- | --- | --- | --- | --- | --- | --- |
| | XSum | ArXiv | PubMed | Yelp | XSum | ArXiv | PubMed | Yelp |
| Likelihood | 0.7430/0.3200 | 0.7074/0.3267 | 0.4892/0.0667 | 0.7811/0.5000 | 0.7414/0.3133 | 0.6990/0.3133 | 0.4824/0.0800 | 0.7880/0.5067 |
| Entropy | 0.3604/0.0133 | 0.5053/0.1333 | 0.5624/0.1333 | 0.2478/0.0200 | 0.3576/0.0133 | 0.5136/0.1133 | 0.5608/0.1467 | 0.2526/0.0067 |
| LogRank | 0.7214/0.2933 | 0.7102/0.3333 | 0.4923/0.0533 | 0.7471/0.4200 | 0.7162/0.2667 | 0.6945/0.3133 | 0.4876/0.0800 | 0.7509/0.4267 |
| LRR | 0.6262/0.1933 | 0.6809/0.2467 | 0.5216/0.0733 | 0.6220/0.2133 | 0.6025/0.1867 | 0.6509/0.2267 | 0.5233/0.0667 | 0.6187/0.1667 |
| NPR | 0.6856/0.3467 | 0.7446/0.5067 | 0.5551/0.1733 | 0.6805/0.3133 | 0.6422/0.3067 | 0.7606/0.5400 | 0.5521/0.1600 | 0.6925/0.3267 |
| DNA-GPT | 0.6714/0.2800 | 0.7064/0.2800 | 0.4890/0.0400 | 0.7631/0.4200 | 0.7035/0.3333 | 0.6975/0.2133 | 0.4805/0.0333 | 0.7411/0.3667 |
| DetectGPT | 0.6388/0.1467 | 0.8321/0.5600 | 0.6058/0.2067 | 0.8348/0.5467 | 0.6296/0.1800 | 0.8205/0.5867 | 0.5874/0.2400 | 0.8095/0.4667 |
| Fast-DetectGPT | 0.7296/0.4200 | 0.8104/0.5267 | 0.6081/0.1467 | 0.6636/0.2800 | 0.7364/0.4200 | 0.8125/0.5600 | 0.5935/0.1133 | 0.6459/0.2733 |
| Fast-Lastde | 0.8173/0.5133 | 0.9227/0.7933 | 0.6116/0.1600 | 0.8450/0.6067 | 0.8281/0.5267 | 0.9358/0.8133 | 0.6204/0.1600 | 0.8292/0.5867 |
| Binoculars | 0.8979/0.7067 | 0.9227/0.7467 | 0.7898/0.4267 | 0.9247/0.7867 | 0.9011/0.7000 | 0.9224/0.7200 | 0.7927/0.4467 | 0.9171/0.7600 |
| MOSAIC | **0.9834**/0.9678 | 0.9786/0.9383 | 0.7961/0.4467 | 0.9741/0.9227 | **0.9826**/0.9556 | 0.9815/0.9533 | 0.7962/0.4467 | 0.9748/0.9367 |
| **SaFT (Ours)** | 0.9826/**1.0000** | **0.9828/0.9933** | **0.9195/0.6933** | **0.9839/0.9600** | 0.9762/**0.9733** | **0.9825/0.9933** | **0.9145/0.7000** | **0.9866/0.9933** |

Table 6: Detailed detection results for text generated by *Gemini-2.5-Flash* and *Gemini-2.5-Pro* models. Results are reported as "AUROC / TPR", where TPR is calculated at 10% FPR. The best results in each column are marked with **bold**.

| Method | Gemini-2.5-Flash | | | | Gemini-2.5-Pro | | | |
| --- | --- | --- | --- | --- | --- | --- | --- | --- |
| | XSum | ArXiv | PubMed | Yelp | XSum | ArXiv | PubMed | Yelp |
| Likelihood | 0.6897/0.2133 | 0.8404/0.5667 | 0.5631/0.1467 | 0.9418/0.8533 | 0.8676/0.5133 | 0.8828/0.6600 | 0.5705/0.1333 | 0.9504/0.8800 |
| Entropy | 0.3799/0.0467 | 0.1996/0.0000 | 0.4680/0.0800 | 0.0984/0.0000 | 0.1688/0.0000 | 0.2334/0.0000 | 0.5336/0.1000 | 0.1107/0.0067 |
| LogRank | 0.6544/0.1800 | 0.7966/0.4067 | 0.5572/0.1400 | 0.8911/0.7200 | 0.8197/0.3800 | 0.8680/0.5733 | 0.5702/0.0800 | 0.9168/0.8000 |
| LRR | 0.5214/0.1067 | 0.5849/0.1533 | 0.5367/0.1267 | 0.6094/0.2067 | 0.5731/0.1400 | 0.7525/0.4267 | 0.5439/0.1133 | 0.6997/0.3600 |
| NPR | 0.5951/0.1933 | 0.5001/0.2200 | 0.5262/0.0933 | 0.5631/0.2067 | 0.3788/0.0400 | 0.5441/0.1667 | 0.5001/0.1600 | 0.5085/0.1800 |
| DNA-GPT | 0.6792/0.2600 | 0.5542/0.1333 | 0.5645/0.0933 | 0.7084/0.3067 | 0.6423/0.1600 | 0.6212/0.1067 | 0.6216/0.1867 | 0.7457/0.3600 |
| DetectGPT | 0.5039/0.0800 | 0.3288/0.0400 | 0.4455/0.0733 | 0.4447/0.0400 | 0.1739/0.0067 | 0.3925/0.0267 | 0.4996/0.0733 | 0.3854/0.0400 |
| Fast-DetectGPT | 0.6486/0.2867 | 0.6475/0.2733 | 0.5763/0.1200 | 0.8586/0.5867 | 0.7472/0.3200 | 0.8349/0.5533 | 0.7053/0.2067 | 0.8736/0.6267 |
| Fast-Lastde | 0.6756/0.2600 | 0.7396/0.4400 | 0.5690/0.0867 | 0.8831/0.7200 | 0.6549/0.3000 | 0.8416/0.5867 | 0.6693/0.2933 | 0.8741/0.5667 |
| Binoculars | 0.7886/0.4733 | 0.8056/0.4200 | 0.7502/0.3600 | 0.9677/0.8933 | 0.8798/0.7067 | 0.9023/0.7100 | 0.8376/**0.5600** | 0.9532/0.8667 |
| MOSAIC | 0.9202/0.7813 | 0.8337/0.4567 | 0.7198/0.2467 | 0.9774/0.9200 | 0.8828/0.6400 | 0.8910/0.6933 | **0.8382/0.5600** | 0.9471/0.8280 |
| **SaFT (Ours)** | **0.9715/0.9800** | **0.9725/0.9733** | **0.8509/0.5400** | **0.9973/1.0000** | **0.9333/0.8333** | **0.9447/0.8667** | 0.8128/0.5133 | **0.9821/0.9867** |

Table 7: Detailed detection results for text generated by *GPT-4o* and *GPT-4.1* models. Results are reported as "AUROC / TPR", where TPR is calculated at 10% FPR. The best results in each column are marked with **bold**.

| Method | GPT-4o | | | | GPT-4.1 | | | |
| --- | --- | --- | --- | --- | --- | --- | --- | --- |
| | XSum | ArXiv | PubMed | Yelp | XSum | ArXiv | PubMed | Yelp |
| Likelihood | 0.9320/0.7400 | 0.9098/0.7533 | 0.5925/0.1533 | 0.9480/0.9200 | 0.9135/0.7000 | 0.9114/0.7733 | 0.6096/0.1667 | 0.8913/0.7200 |
| Entropy | 0.0841/0.0000 | 0.2359/0.0200 | 0.5162/0.0867 | 0.0772/0.0200 | 0.1191/0.0000 | 0.1367/0.0000 | 0.4963/0.0800 | 0.1461/0.0000 |
| LogRank | 0.8807/0.5800 | 0.8889/0.6867 | 0.5909/0.1467 | 0.9255/0.8467 | 0.8643/0.5400 | 0.8800/0.6400 | 0.5997/0.1467 | 0.8503/0.5600 |
| LRR | 0.5902/0.1467 | 0.7218/0.3000 | 0.5741/0.1600 | 0.7209/0.3200 | 0.6009/0.2133 | 0.6751/0.2067 | 0.5547/0.0933 | 0.6580/0.2600 |
| NPR | 0.3424/0.0133 | 0.6765/0.3867 | 0.5253/0.1333 | 0.6263/0.2933 | 0.4421/0.0667 | 0.5220/0.2667 | 0.5355/0.1400 | 0.7074/0.3867 |
| DNA-GPT | 0.6931/0.3667 | 0.7909/0.4200 | 0.6075/0.0733 | 0.8636/0.6600 | 0.6860/0.2800 | 0.6824/0.1733 | 0.6301/0.1000 | 0.8449/0.5733 |
| DetectGPT | 0.1881/0.0000 | 0.5336/0.1133 | 0.5069/0.0800 | 0.6336/0.1400 | 0.2099/0.0000 | 0.3534/0.0267 | 0.5024/0.0733 | 0.7592/0.4333 |
| Fast-DetectGPT | 0.7454/0.3800 | 0.8680/0.6267 | 0.7163/0.2867 | 0.8834/0.7067 | 0.7218/0.3267 | 0.7627/0.3600 | 0.6971/0.2467 | 0.7803/0.4467 |
| Fast-Lastde | 0.6246/0.2067 | 0.9003/0.7333 | 0.6587/0.1733 | 0.8938/0.7533 | 0.7459/0.3733 | 0.8603/0.6333 | 0.6448/0.1600 | 0.8788/0.6533 |
| Binoculars | 0.9046/0.7267 | 0.9684/0.8867 | 0.8900/0.7067 | **0.9740**/0.9333 | 0.8666/0.6600 | 0.9088/0.6933 | 0.8807/0.6533 | 0.9419/0.8133 |
| MOSAIC | 0.9678/0.9156 | 0.9620/0.8867 | 0.8593/0.5489 | 0.9714/0.9467 | 0.9358/0.8560 | 0.9266/0.7667 | 0.8437/0.4733 | 0.9769/0.9237 |
| **SaFT (Ours)** | **0.9823/1.0000** | **0.9892/1.0000** | **0.9463/0.8400** | 0.9632/**0.9667** | **0.9828/0.9867** | **0.9901/1.0000** | **0.9560/0.8533** | **0.9893/0.9867** |

