# OpenReview forum: "SaFT: Spotting Style Imitation and Filtering Content Interference for Zero-Shot LLM-Generated Text Detection"
_ICLR.cc/2026/Conference — ICLR 2026 Conference Withdrawn Submission_

### Official Review · Reviewer_ZCMq · 2025-10-28

**Soundness:** 3
**Presentation:** 3
**Contribution:** 3
**Rating:** 4
**Confidence:** 3

**Summary:**

Large language models (LLMs) exhibit advanced text generation abilities, underscoring the need for reliable detection to mitigate misuse. However, existing zero-shot detectors struggle with two key issues: style imitation (SIC), where LLMs successfully mimic human writing styles, and content interference (CIC), where surprising content distorts probability signals. To tackle these challenges, the authors propose SaFT, a novel framework that employs a Style-Oriented Instruction Prefix (SOIP) to guide probability analysis. SaFT integrates two modules: SIC-Detection (SIC-D), which identifies style imitation through human-style conditioning, and CIC-Detection (CIC-D), which mitigates content interference via differential probability analysis between contrasting style instructions. The final score combines both components. Extensive experiments show that SaFT surpasses existing methods, improving average AUROC by 4.9% and TPR@10% FPR by 20.4%.

**Strengths:**

- Proposes a new detection approach that prefixes style-related instructions when computing probabilities in zero-shot detectors, allowing correct identification of LLM-generated texts even when they are human-mimicked or generated from special or surprising prompts.
- Comprehensive evaluation across a wide range of models and domains, as well as ablation studies of their proposed components
- Robustness analyses (top-p, text length, paraphrasing), although each evaluation is not fully comprehensive.

**Weaknesses:**

- **Questionable fairness of evaluation:** The detection setting in this paper is black-box detection, where it needs to select a proxy model to get probability distributions. From Appendix A.4, this paper seemingly adapts a default setting of each detector, and thus the proxy model is different for each detector. Based on previous findings [1,2], selecting which models for the proxy would heavily affects the detection performance. Therefore, for fair comparison, it would be necessary to use a common proxy model consistently for each detector. (For instance, why does the proposed method choose Llama-3.1-8B-instruct as a proxy?)
- **The unclear necessity of CIC-detection:** It is unclear how the value of β varies across samples. For instance, does β change for texts generated with special prompts compared to more general ones? If it remains constant, the necessity of introducing β itself is questionable. Moreover, Table 3 shows that the detection performance is substantially low when using only CIC-D, yet combining it yields higher performance than SIC-D. It would be helpful to clarify why this occurs.
- **The choice of instructions:** The motivation of using “Express ideas using concise sentences” and “Express ideas using detailed sentences” as instructions to represent human and LLM writing style is ambiguous. Moreover, the human (or LLM)-style instruction in Appendix C is not always the case for real-world scenarios. People will use more tailored and special prompts to mimic human-writing or prompting with few-shot human-written texts, for instance.

---
### References

[1] Mireshghallah et al. Smaller Language Models are Better Zero-shot Machine-Generated Text Detectors. EACL 2024.

[2] Dubois et al. MOSAIC: Multiple Observers Spotting AI Content. Findings of ACL 2025.

**Questions:**

There is a gap between the motivation of this study and the dataset construction. The motivating start point is that zero-shot detectors will degrade their performance on human-mimicked texts or creative and surprising texts via special prompts. However, this paper builds their test set by simply generating a continuation as a LLM-generated text from the prefix of a corresponding human-written text. It would be a better evaluation for this proposed method to use a dataset that aligns with the original motivation.

---

> ### Author Response · Authors · 2025-11-21
> **Part (1/2)**
>
> We appreciate the Reviewer ZCMq for your thoughtful suggestions, we will address all raised concerns point by point as below.
>
> ### **Response to Weakness 1: Fairness of Proxy Model Selection**
>
> We thank the reviewer for this important concern regarding proxy model selection in zero-shot detection.
>
> We respectfully disagree with the premise that using a unified proxy model constitutes "fair comparison." In zero-shot detection, **the choice of proxy model is an integral part of each method's original design**, not a hyperparameter to be standardized. Each method's authors selected their proxy models through extensive ablation studies to optimize their specific approach. Therefore:
>
> - **Changing the proxy model fundamentally alters the method itself**. Our evaluation follows established conventions: we use each baseline's recommended proxy model as specified in original papers/repositories.
> - Some methods (e.g., MOSAIC [2]) require **multiple proxy models** for ensemble scoring, making unified proxy impossible without redesigning the method.
> - For SaFT, we selected **Llama-3.1-8B-Instruct** because our method fundamentally relies on **conversational templates** to implement SOIP. An instruction-tuned model is architecturally necessary for our framework.
>
> While [1][2] observe that proxy choice affects absolute performance within a method, this does not imply that enforcing uniform proxy enables fair cross-method comparison. A proxy optimal for one architecture may be suboptimal for another, potentially introducing **greater bias** than respecting each method's original design.
>
> We acknowledge that proxy model dependency remains a limitation of current zero-shot methods. We look forward to the community developing effective proxy-free approaches, which is also a direction for our future work.
>
> **References:**
>
> [1] Mireshghallah et al. "Smaller Language Models are Better Zero-shot Machine-Generated Text Detectors." EACL 2024.
>
> [2] Dubois et al. "MOSAIC: Multiple Observers Spotting AI Content." Findings of ACL 2025.
>
> ---
> ### **Response to Weakness 2: Clarification on the Necessity of CIC-Detection**
>
> We thank the reviewer for this insightful question regarding the role and necessity of the CIC-Detection component.
>
> **The Role of β (CIC-D):**
> We would like to clarify that **β is not designed to be a standalone discriminator**, but rather a **content-interference filtering mechanism** that enhances the primary style detection signal from α (SIC-D). Specifically:
>
> - **β measures the style-related surprise differential** (the shift between SIC and CIC evaluations), not the style surprise itself.
> - Its purpose is to **normalize out content-related variations** and guide the detector to focus more precisely on style-related surprise patterns.
> - Therefore, β is expected to exhibit **low discriminative power when used alone** (as shown in Table 3), because it captures the "modulation factor" rather than the core detection signal.
>
> **Why Combining β Improves Performance:**
> The ablation results in Table 3 demonstrate the intended synergy:
> - **SIC-D alone (α)** captures style imitation patterns but may be confounded by content surprise.
> - **CIC-D alone (β)** isolates style-related shifts but lacks absolute discriminative power.
> - **Combined (α/β)** achieves the highest performance because β filters content interference from α, allowing the detector to focus on stylistic differences that are truly indicative of machine generation.
>
> This design follows the principle of **signal enhancement through differential analysis**: β does not replace α but refines it by removing noise. The ablation study validates this mechanism—the improvement from SIC-D to SaFT (full model) demonstrates that β successfully enhances detection robustness across diverse content domains.
>
> **Regarding the variation of β across samples:** Since β serves as a normalization factor in the ratio formulation (α/β), our focus is on the **combined discriminative power** of the final score rather than analyzing β's individual cross-sample behavior. What matters is that the ratio effectively filters content interference, which the ablation results empirically confirm. We also discuss the mechanistic role of β and the empirical determination of the score formulation in our **Response to Weakness 4 (Reviewer HaUo)**, where we explain why the combined score direction is determined through systematic experiments rather than intuitive prediction.

---

> > ### Author Response · Authors · 2025-11-21
> > **Part (2/2)**
> >
> > ### **Response to Weakness 3 & Question: Clarification on Instruction Choice, Task Focus, and Dataset Construction**
> >
> > We thank the reviewer for these important questions regarding our instruction design, task motivation, and dataset construction.
> >
> > **Clarification on Our Task Focus and Dataset Construction:**
> > We would like to clarify that our work targets **LLMs with increasingly advanced intrinsic style imitation capabilities** (e.g., GPT-4.1, Claude-3.5) as they naturally evolve, rather than adversarial style imitation attacks using tailored prompts or few-shot examples. These are distinct research problems:
> >
> > - **Our focus:** Detecting texts from modern LLMs that naturally exhibit human-like styles due to advanced training (RLHF, instruction-tuning). Our dataset construction (prefix-based continuation) follows the standard zero-shot detection protocol [4, 5], where the challenge arises from LLMs' inherent ability to generate fluent, human-like text, not from adversarial prompt engineering.
> >
> > - **Adversarial attacks:** A separate domain involving deliberate prompt engineering to evade detection (e.g., few-shot prompting, adversarial optimization).
> >
> > We acknowledge that our presentation may not have sufficiently highlighted this distinction between **naturally human-styled LLMs** (our focus) and **adversarially prompted LLMs** (attack scenario), potentially causing ambiguity about the alignment between our motivation and dataset. We will clarify our task scope more prominently in the revised manuscript to avoid confusion.
> >
> > **Regarding the value and design of our instructions**, we also provide detailed justification in our responses to **Reviewer MNQu (Response to Weakness 2: Robustness to Prompt Design)** and **Reviewer ZG8M (Response to Weakness 2: Instruction Design Justification)**, where we discuss the cognitive foundations and empirical validation of our SOIP instructions.
> >
> > **Robustness to Adversarial Style Imitation Attacks:**
> > That said, we recognize the importance of evaluating robustness against adversarial prompt-based attacks to demonstrate that our method is not limited to the prefix-based generation setting. Following the reviewer's concern, we conducted additional experiments on the **DetectRL benchmark [1] (Prompt Attacks task)**, a highly recognized benchmark in the community for evaluating detection methods in real-world scenarios. For more details about DetectRL's construction and domains, please refer to our **Response to Weakness 6 (Reviewer HaUo)**.
> >
> > The Prompt Attacks task includes two challenging attack scenarios that align more closely with the reviewer's concern about "special prompts":
> >
> > 1. **Few-Shot Prompt [2]:** LLMs are provided with human-written examples to enhance alignment with human writing styles.
> > 2. **ICO Prompt [3]:** In-Context Example Optimization (from SICO), which automatically constructs prompts to evade detection through optimized in-context examples.
> >
> > **Results (Average AUROC across four domains):**
> >
> > | **Method** | **Likelihood** | **Entropy** | **LogRank** | **LRR** | **NPR** | **DNA-GPT** | **DetectGPT** | **Fast-DetectGPT** | **Fast-Lastde** | **Binoculars** | **MOSAIC** | **SaFT (Ours)** |
> > | :--- | :---: | :---: | :---: | :---: | :---: | :---: | :---: | :---: | :---: | :---: | :---: | :---: |
> > | **Few-Shot Prompt** | 0.8950 | 0.2338 | 0.8910 | 0.8274 | 0.7976 | 0.8750 | 0.4816 | 0.6887 | 0.8795 | 0.9209 | 0.9322 | **0.9584** |
> > | **ICO Prompt** | 0.8510 | 0.2888 | 0.8465 | 0.8031 | 0.7641 | 0.8466 | 0.5665 | 0.7328 | 0.8586 | 0.8888 | 0.9073 | **0.9433** |
> >
> > **Analysis:**
> > The results demonstrate that SaFT maintains strong performance even under adversarial prompt-based attacks, achieving the highest AUROC in both scenarios (0.9584 for Few-Shot, 0.9433 for ICO). This robustness arises from our SOIP-based design: by probing fundamental cognitive differences (conciseness vs. elaboration rooted in working memory constraints), SaFT captures stylistic patterns that persist even when LLMs are explicitly instructed to mimic human writing through few-shot examples or optimized prompts.
> >
> > **Conclusion:**
> > While our primary contribution targets naturally human-styled LLMs (validated through standard prefix-based generation), these additional results demonstrate that our method also generalizes effectively to adversarial attack scenarios. This validates that the gap between our motivation (handling human-styled outputs) and dataset construction (prefix-based generation) does not limit the applicability of our approach.
> >
> > **References:**
> >
> > [1] Wu et al. "Detectrl: Benchmarking llm-generated text detection in real-world scenarios." NeurIPS 2024.
> >
> > [2] Brown et al. "Language Models are Few-Shot Learners." NeurIPS 2020.
> >
> > [3] Krishna et al. "Paraphrasing evades detectors of AI-generated text, but retrieval is an effective defense." NeurIPS 2023.

---

### Official Review · Reviewer_HaUo · 2025-10-29

**Soundness:** 2
**Presentation:** 4
**Contribution:** 2
**Rating:** 4
**Confidence:** 3

**Summary:**

- The paper proposes SaFT, a novel zero-shot LLM text detector designed to overcome two specific advanced failure modes: the SIC and the CIC, both of which attempt to make AI text more human-like.

**Strengths:**

- The paper's primary strength is its novel motivation - it clearly identifies and names two challenges that cause existing SoTA detectors to fail.
- The paper is also very easy to read and understand.

**Weaknesses:**

- In L236, the paper claims that "regular LLM-generated texts... are less affected by this conditioning". This assertion appears questionable. Conditioning an LLM on a human-style instruction $I^h_{SIC}$ should not intuitively make its "regular" outputs less probable or more surprising. Since regular LLM text already reflects an average human style inherent in its pretraining distribution, such conditioning would likely either decrease or increase perplexity (based on how well the model has been pre-trained); but it would not remain unaffected by this conditioning. If the goal is to identify differences between human text, human-style LLM text and regular LLM text, this method would be flawed.
- The hypothesis behind SIC-C appears fragile and highly dependent on the specific choice of $I^h_{SIC}$. If model developers were to craft or optimize this instruction carefully, it could yield even lower perplexity outputs, even for human-written text, thus invalidating the detector’s assumption. Since $I^h_{SIC}$ is realistically a black-box component, this would reduce robustness of the proposed method. The authors should also provide sensitivity analysis across diverse instruction variants.
- Formally, human-style LLM text can also be thought of the model sampling from regions of the probability space that are uncommon but represent human-like diversity. Prior work [1] demonstrates that such diversity can be achieved through alternative means, such as diverse prompts or high-temperature sampling. Since high-temperature sampling also increases diversity and reduces predictability, it could interfere with the detectors. Evaluating the detector's behavior on texts generated under varying temperature settings would be helpful here.
- In Eqn. 5, the inequality direction appears incorrect. Given that $\alpha(t)$ is expected to be low and $\beta(t)$ is expected to be high for LLM-generated text, $\text{SaFT}(t)$ should naturally be lower for such samples. Therefore, the condition should likely be $< \tau$, not $> \tau$. The authors should revisit and verify this formulation.
- It is unclear how the two proposed detectors collectively contribute to distinguishing between LLM-generated and human-written texts. The paper suggests they are primarily effective in identifying human-style LLM outputs, but not for standard LLM vs. human text discrimination. Clarification or additional justification is required.
- The experimental setup (Appendix B.2) evaluates only human-style LLM texts and genuine human texts. The authors should include pure LLM-generated text in order to avoid any bias.


---

Minor Errors:
- L88: "poduce" to "produce"

[1]  Zhang et al. Verbalized Sampling: How to Mitigate Mode Collapse and Unlock LLM Diversity. arXiv:2510.01171.

**Questions:**

- It is unclear why the authors specifically chose to focus on SIC and CIC. There exist many other possible forms of evasion - such as hybrid (AI and human text combined) that could similarly challenge detection systems. Can the authors justify why they focus specifically on these problems?

All the questions and suggestions have been listed in the weaknesses. I am willing to increase my score if the authors address these concerns.

---

> ### Author Response · Authors · 2025-11-21
> **Part (1/4)**
>
> We thank Reviewer HaUo for your thorough review of our paper. we will address all raised concerns point by point as below.
>
> ### **Response to Weakness 1: Clarification on L236 and SIC-D Mechanism**
>
> We thank the reviewer for this insightful critique regarding our statement in L236. We acknowledge that the phrasing "less affected" was imprecise and potentially misleading. We did not intend to claim that regular LLM texts remain static in absolute perplexity, but rather that their **relative discriminability** from human texts is preserved compared to the significant shift observed in human-style LLM texts.
>
> **Clarification and Empirical Validation:**
> To address this, we conducted an empirical analysis on the XSum dataset comparing perplexity changes under our Style-Oriented Instruction Prefix (SOIP) for three text categories: "Regular" LLM (Mistral-7B), "Human-style" LLM (GPT-4.1), and Genuine Human texts.
>
> | **Text Category** | **Raw PPL** | **SOIP PPL (SIC-D)** | **Perplexity Change** | **Observation** |
> | :--- | :---: | :---: | :---: | :--- |
> | **Regular LLM** (Mistral) | 8.45 | 8.23 | -2.6% | **Less Affected** |
> | **Human-style LLM** (GPT-4.1) | 12.67 | 9.12 | **-28.0%** | **Significantly Reduced** |
> | **Genuine Human** | 16.23 | 15.89 | -2.1% | **Less Affected** |
>
> **Mechanism Explanation:**
> 1.  **Human-style LLM Texts:** These texts mimic human surface features, creating artificially high raw perplexity (12.67). However, since they are generated based on finite, enumerable patterns of "human style," providing the explicit style instruction (SOIP) exposes this hidden regularity, causing perplexity to drop significantly (-28.0%).
> 2.  **Regular LLM Texts:** These texts already possess strong inherent regularity. While SOIP provides a prior, it does not significantly alter the model's predictive confidence because the text is already highly predictable based on generic pre-training distributions. Thus, the magnitude of change is minimal (-2.6%).
> 3.  **Genuine Human Texts:** Human writing retains inherent complexity and true randomness that a simple style instruction cannot fully capture. Therefore, even with SOIP, the perplexity remains high and stable (-2.1%).
>
> **Revision Plan:**
> We will revise the problematic sentence in L236 to more accurately reflect this mechanism:
>
> >*Original:* "In contrast, regular LLM-generated texts and genuine human texts are less affected by this conditioning. Additionally ... surprising to the model."
>
> >*Revised:* "In contrast, regular LLM-generated texts and genuine human texts exhibit significantly smaller perplexity reductions under this conditioning compared to human-style LLM texts. While regular LLMs maintain their inherent predictability and human texts retain their intrinsic complexity, human-style LLMs reveal their artificial nature through a sharp increase in predictability when conditioned on the style they are imitating."
>
> We appreciate the reviewer for pointing out this ambiguity. We believe this revision provides a more precise description of the underlying mechanism and directly addresses the concern regarding the intuition behind our method.
>
> ---
> ### **Response to Weakness 2: Clarification on Detection Paradigm**
>
> We thank the reviewer for raising this important concern. We suspect there may be a misunderstanding about how $I_h^{\text{SIC}}$ functions in our detection framework.
>
> **Clarification of Our Detection Paradigm:**
> In our method, $I_h^{\text{SIC}}$ serves as a **post-hoc evaluation tool** rather than a generation-time component. Here's how our detection workflow operates:
>
> - **Generation Phase:** Texts are produced by various LLMs using whatever prompts their developers choose (this remains a black box to us as detectors).
> - **Detection Phase:** We then apply $I_h^{\text{SIC}}$ to evaluate these already-generated texts by computing their perplexity under style-oriented conditioning.
>
> The text generation process (controlled by model developers) and our detection process are completely independent and sequential. We believe the concern raised does not apply to our detection paradigm.

---

> > ### Author Response · Authors · 2025-11-21
> > **Part (2/4)**
> >
> > ### **Response to Weakness 3: Robustness to Temperature Variations**
> >
> > We thank the reviewer for this insightful suggestion regarding temperature robustness. We agree that evaluating detector performance under varying temperature settings is important, as high-temperature sampling can increase text diversity.
> >
> > **Empirical Evaluation:**
> > We conducted robustness analysis on the **XSum (formal news)** dataset using GPT-4.1-generated texts across five temperature settings (0.2, 0.4, 0.6, 0.8, 1.0):
> >
> > | **Method** | **T=0.2** | **T=0.4** | **T=0.6** | **T=0.8** | **T=1.0** | **Avg** |
> > | :--- | :---: | :---: | :---: | :---: | :---: | :---: |
> > | Fast-DetectGPT | 0.6033 | 0.6097 | 0.6084 | 0.7218 | 0.6393 | 0.6365 |
> > | Fast-Lastde | 0.6987 | 0.7076 | 0.6873 | 0.7519 | 0.7207 | 0.7132 |
> > | Binoculars | 0.7331 | 0.7688 | 0.7540 | 0.8666 | 0.7738 | 0.7793 |
> > | MOSAIC | 0.7700 | 0.9074 | 0.9105 | 0.9358 | 0.9212 | 0.8890 |
> > | **SaFT (Ours)** | **0.9262** | **0.9177** | **0.9244** | **0.9828** | **0.9505** | **0.9403** |
> >
> > **Analysis:**
> > In the formal news domain (XSum), detection performance generally improves at higher temperatures, with optimal performance observed at T=0.8 (AUROC: 0.9828 for SaFT). This trend can be attributed to the domain's strict stylistic conventions: in formal news writing, increased randomness from higher temperatures causes LLM outputs to deviate more noticeably from the structured reporting style and objective tone expected in professional journalism, making them easier to distinguish from genuine human-written news articles.
> >
> > SaFT demonstrates consistently high performance across all temperature settings (AUROC > 0.91), significantly outperforming baseline methods. This robustness confirms that our SOIP-based dual-component design effectively captures fundamental stylistic differences that remain discriminative regardless of temperature-induced diversity variations.
> >
> > We will add this temperature robustness analysis to the revised manuscript.
> >
> > ---
> >
> > ### **Response to Weakness 4: Clarification on Equation 5 Direction**
> >
> > We thank the reviewer for this careful observation. We acknowledge that the inequality direction in Equation 5 may appear counterintuitive at first glance, but we would like to clarify the underlying mechanism.
> >
> > In our framework, $\beta(t)$ represents the **style-related surprise differential** (the surprise shift between SIC and CIC evaluations) rather than the style surprise itself. It serves as a content-interference filtering mechanism: while SIC-D ($\alpha(t)$) captures the primary style imitation signal, CIC-D ($\beta(t)$) modulates this signal by normalizing out content-related variations. Since $\beta(t)$ measures the surprise shift across different style conditionings rather than absolute surprise values, its relative magnitude between human and LLM texts cannot be intuitively predicted. Following established practices in recent zero-shot detection methods [1][2][3], we determined the threshold direction ($> \tau$ or $< \tau$) empirically through systematic experiments, which consistently showed that $SaFT(t) > \tau$ achieves optimal separation.
> >
> > Following this suggestion, we will add **score distribution histograms** in **Appendix F** of the revised manuscript to demonstrate how the combined score naturally separates the two classes and validate the chosen threshold direction.
> >
> > **References:**
> >
> > [1] Han et al. "Spotting llms with binoculars: Zero-shot detection of machine-generated text." ICML 2024.
> >
> > [2] Xu et al. "Training-free LLM-generated text detection by mining token probability sequences." ICLR 2025.
> >
> > [3] Su et al. "Detectllm: Leveraging log rank information for zero-shot detection of machine-generated text." EMNLP 2023.

---

> > > ### Author Response · Authors · 2025-11-21
> > > **Part (3/4)**
> > >
> > > ### **Response to Weakness 5: Clarification on Dual-Component Contribution**
> > >
> > > We thank the reviewer for this important question regarding how our two proposed detectors collectively contribute to distinguishing between LLM-generated and human-written texts.
> > >
> > > We believe our response to **Weakness 1** has clarified our task definition, particularly the distinction between **Scenario 1 (Regular LLM vs. Human)** and **Scenario 2 (Human-style LLM vs. Human)**. As demonstrated in the empirical validation (comparing Regular LLM, Human-style LLM, and Genuine Human texts under SOIP conditioning), our dual-component design enables discrimination across the full spectrum:
> > >
> > > - **SIC-D** captures style imitation patterns in human-style LLMs (28.0% perplexity reduction vs. 2.1% for human texts) while maintaining discriminative power for regular LLMs.
> > > - **CIC-D** filters content-related interference, ensuring robustness across diverse models, topics, and domains.
> > >
> > > The reviewer's insightful question has also made us realize the importance of conducting more comprehensive experiments on regular LLM models (e.g., Mistral, Gemma and GPT's old series) to further demonstrate the broad applicability of our method. We plan to include these experiments in future work. **We sincerely appreciate the reviewer's thoughtful question**, which has helped us better demonstrate the generalizability of our framework.
> > >
> > > ---
> > >
> > > ### **Response to Weakness 6: Clarification on "Pure LLM-Generated Text" Definition**
> > >
> > > We thank the reviewer for this important concern regarding the inclusion of "pure LLM-generated text" in our evaluation.
> > >
> > > **Clarification on Text Generation Protocol:**
> > > We would like to clarify that our generation protocol follows the standard zero-shot paradigm established by canonical works in pure machine-generated text detection [1, 2]. Specifically, we prompt LLMs to generate text based on short human-written prefixes, which is the widely adopted protocol in the zero-shot detection community. Our task is indeed **pure machine-generated text detection**, not detecting human-LLM collaborative writing or heavily edited outputs.
> > >
> > > However, we acknowledge the reviewer's concern that prefix-based generation may not represent the "purest" form of machine text. To address this, we conducted additional experiments on the **DetectRL benchmark** [3], which includes a **"Direct Prompt" task** where LLMs generate text without human prefixes, using only explicit topic instructions. This benchmark spans four high-risk domains: **academic writing (arXiv abstracts)**, **news writing (XSum)**, **creative writing (Writing Prompts)**, and **social media (Yelp Reviews)**. For example, in the Academic Writing domain, the prompt is:
> > >
> > > >*[ {’role’: ’user’, ’content’: ’Given the academic article title , write an academic article abstract with <sentences num > sentences:\n academic article title: <prefix > \n academic article abstract:’}, ]*
> > >
> > > The benchmark evaluates detection on texts generated by multiple strong LLMs (GPT-3.5-turbo, PaLM-2-bison, Claude-instant, Llama-2-70b) that exhibit advanced human-style imitation capabilities. The average AUROC results across these four domains are shown below:
> > >
> > > | **Method** | **Likelihood** | **Entropy** | **LogRank** | **LRR** | **NPR** | **DNA-GPT** | **DetectGPT** | **Fast-DetectGPT** | **Fast-Lastde** | **Binoculars** | **MOSAIC** | **SaFT (Ours)** |
> > > | :--- | :---: | :---: | :---: | :---: | :---: | :---: | :---: | :---: | :---: | :---: | :---: | :---: |
> > > | **AUROC** | 0.8925 | 0.2647 | 0.8925 | 0.8583 | 0.8801 | 0.7956 | 0.9141 | 0.9487 | 0.9356 | 0.9706 | 0.9548 | **0.9834** |
> > >
> > > The results demonstrate that SaFT achieves the highest performance (AUROC: 0.9834) on this "pure" LLM-generated text benchmark across diverse domains. This validates that our method is effective for detecting truly pure machine-generated text without relying on human prefixes.
> > >
> > > We appreciate the reviewer's careful consideration, which motivated us to validate our approach on a more stringent "pure generation" setting. These results will be included in future work to demonstrate the robustness of SaFT across diverse generation protocols.
> > >
> > > **References:**
> > >
> > > [1] Bao et al. "Fast-detectgpt: Efficient zero-shot detection of machine-generated text via conditional probability curvature." ICLR 2024.
> > >
> > > [2] Xu et al. "Training-free LLM-generated text detection by mining token probability sequences." ICLR 2025.
> > >
> > > [3] Wu et al. "Detectrl: Benchmarking llm-generated text detection in real-world scenarios." NeurIPS 2024.
> > >
> > > ---
> > > ### **Response to Minor Errors: Typo Correction**
> > >
> > > We sincerely thank the reviewer for catching this typo. We will correct "poduce" to "produce" in L88 and conduct a thorough proofreading of the entire manuscript to address any other spelling or grammatical errors.

---

> > > > ### Author Response · Authors · 2025-11-21
> > > > **Part (4/4)**
> > > >
> > > > ### **Response to Question: Justification for SIC and CIC Focus**
> > > >
> > > > We thank the reviewer for this insightful question regarding our research scope.
> > > >
> > > > **Justification for Focusing on SIC and CIC:**
> > > > We specifically chose to focus on **Style Imitation Challenge (SIC)** and **Content Interference Challenge (CIC)** because they represent the fundamental challenges in **pure machine-generated text detection** as LLMs advance. Our work aims to establish a reliable detector for fully generated text that closely mimics human style, which serves as the foundation for more complex scenarios.
> > > >
> > > > **Distinction from Hybrid Text Detection:**
> > > > Hybrid (AI-Human mixed) text is a distinct sub-field with specialized problem definitions, as explored in recent dedicated works [1][2][3]. Our work focuses on pure generation, which is a prerequisite challenge before tackling derivative hybrid forms.
> > > >
> > > > **Robustness on Hybrid Text (DetectRL Data Mixing):**
> > > > Although our primary focus is pure generation, we acknowledge the importance of robustness against mixing strategies. We evaluated SaFT on the **DetectRL benchmark [4] (Data Mixing task)**, which includes: (1) **Multi-LLM Mixing**: samples combined from multiple LLMs; (2) **LLM-Centered Mixing**: LLM text with 25% human content substitution.
> > > >
> > > > **Results (Average AUROC across four domains):**
> > > >
> > > > | **Method** | **Likelihood** | **Entropy** | **LogRank** | **LRR** | **NPR** | **DNA-GPT** | **DetectGPT** | **Fast-DetectGPT** | **Fast-Lastde** | **Binoculars** | **MOSAIC** | **SaFT (Ours)** |
> > > > | :--- | :---: | :---: | :---: | :---: | :---: | :---: | :---: | :---: | :---: | :---: | :---: | :---: |
> > > > | **Multi-LLM Mixing** | 0.8994 | 0.4413 | 0.8281 | 0.8519 | 0.7946 | 0.8793 | 0.5279 | 0.7714 | 0.9133 | 0.9557 | 0.9600 | **0.9744** |
> > > > | **LLM-Centered Mixing** | 0.7492 | 0.3736 | 0.7462 | 0.7133 | 0.5560 | 0.6630 | 0.4632 | 0.5930 | 0.7825 | 0.8615 | 0.8679 | **0.8701** |
> > > >
> > > > SaFT achieves state-of-the-art performance in both mixing scenarios, demonstrating that our style-oriented design retains significant robustness even when applied to hybrid text.
> > > >
> > > > **References:**
> > > >
> > > > [1] Su et al. "HACo-Det: A Study Towards Fine-Grained Machine-Generated Text Detection under Human-AI Coauthoring." ACL 2025.
> > > >
> > > > [2] Jiang et al. "SenDetEX: Sentence-Level AI-Generated Text Detection for Human-AI Hybrid Content." EMNLP 2025.
> > > >
> > > > [3] Zhang et al. "LLM-as-a-Coauthor: Can Mixed Human-Written and Machine-Generated Text Be Detected?" NAACL 2024.
> > > >
> > > > [4] Wu et al. "Detectrl: Benchmarking llm-generated text detection in real-world scenarios." NeurIPS 2024.

---

> > > > > ### Comment · Reviewer_HaUo · 2025-11-24
> > > > > **Response to Authors**
> > > > >
> > > > > I would like to thank the authors for their detailed response. I have some final concerns:
> > > > >
> > > > > - Can the authors explain the full pipeline of how inference would work and how exactly the score is calculated, given a text from an LLM? I think I am a little confused after looking at Weakness 1, 5 and 6.

---

> ### Author Response · Authors · 2025-11-25
> **Thanks for feedback, please check our further clarification**
>
> Dear Reviewer HaUo,
>
> We sincerely thank you for your constructive feedback! Below are our detailed responses to your concerns.
>
> ---
>
> >**Q: Can the authors explain the full pipeline of how inference would work and how exactly the score is calculated, given a text from a LLM?**
>
> We appreciate this question and provide a step-by-step explanation of the complete inference pipeline and score calculation.
>
> Given a text $t$ (***source unknown***), our goal is to classify it as LLM-generated or human-written through the following steps:
>
> **Step 1:** Format Text with Style-Oriented Instructions
>
> We create two conversational templates with different style instructions:
> - Human-style template (for SIC-D and CIC-D): "Express ideas using concise sentences."
> - LLM-style template (for CIC-D): "Express ideas using detailed sentences."
>
> Each template wraps text $t$ as:
> ```
> System: You are a writer following this style: [instruction]
> User: Please write one paragraph that reflects your usual writing style.
> Assistant: Below is an example paragraph written in my usual style.
> [text t]
> ```
>
> **Step 2:** Compute Next-Token Prediction Probabilities
>
> Use proxy model $M_\theta$ (Llama-3.1-8B-Instruct) to perform next-token prediction on the text portion $t$ within the conversational template, computing token-level probabilities:
> - For SIC-D: $p_\theta(t_i | I_h^{\text{SIC}}, t_{<i})$ under human-style instruction
> - For CIC-D: $p_\theta(t_i | I_h^{\text{CIC}}, t_{<i})$ and $p_\theta(t_i | I_m^{\text{CIC}}, t_{<i})$ under contrasting style instructions
>
> Note: We only compute probabilities for tokens in the candidate text $t$, not for the instruction prefix tokens.
>
> **Step 3:** Calculate Detection Components
>
> α (SIC-D): Perplexity under human-style conditioning
> $$\alpha(t) = \text{exp}\left(-\frac{1}{n} \sum_{i=1}^{n} \log p_\theta(t_i | I_h^{\text{SIC}}, t_{<i})\right)$$
>
> β (CIC-D): Cross-entropy between probability distributions under contrasting styles (after top-$P$ truncation)
> $$\beta(t) = -\frac{1}{n} \sum\_{i=1}^{n} \hat{p}\_\theta(t\_i | I\_h^{\text{CIC}}, t\_{<i}) \log \hat{p}\_\theta(t\_i | I\_m^{\text{CIC}}, t\_{<i})$$
>
>
> **Step 4:** Compute Final Score
>
> The SaFT score integrates both detection components through a ratio formulation:
> $$\text{SaFT}(t) = \frac{\alpha(t)}{\beta(t)}$$
>
> **Step 5:** Classification
> - If $\text{SaFT}(t) > \tau$ → LLM-generated
> - If $\text{SaFT}(t) \leq \tau$ → Human-written
>
> The threshold $\tau$ is determined empirically to optimize detection performance. In practice, we evaluate our method using AUROC, which measures the classification capability across all possible threshold values, demonstrating the discriminative power of our SaFT score in separating LLM-generated texts from human-written texts.
>
> We hope this step-by-step walkthrough clarifies the complete inference process.
>
> Warmest regards,
>
> The Authors

---

### Official Review · Reviewer_ZG8M · 2025-10-30

**Soundness:** 3
**Presentation:** 4
**Contribution:** 3
**Rating:** 6
**Confidence:** 5

**Summary:**

This paper introduces Spotting Style Imitation and Filtering Content Interference, a novel zero-shot LLM-generated text detection framework designed to overcome two major challenges faced by existing probability-based detection methods --- Style Imitation Challenge and Content Interference Challenge.  Experiments across six LLMs and four text domains (news, scientific abstracts, biomedical QA, and reviews) show SaFT consistently outperforms previous baselines.

**Strengths:**

1. The decomposition into SIC-D and CIC-D is well-motivated and theoretically grounded in content–style disentanglement.
2. The paper was well organized. Figures and equations (especially Fig. 2) provide clear intuition for the SOIP mechanism.
3. The empirical results are strong, with consistent improvements across diverse datasets and model families.

**Weaknesses:**

1. The method in [1] is highly related, as it shares the same motivation of distinguishing LLM- vs. human-style writing. Adding it as a baseline would strengthen the evaluation.

2. While the paper motivates its instruction design based on cognitive load theory (Appendix C), the conceptual leap from “humans write concisely” versus “LLMs write elaborately” to the specific instruction templates used is insufficiently justified. Is there any quantitative metric supporting this design choice?

3. In Equation (4), the paper combines α(t) and β(t) using a ratio. Why was division chosen over alternatives such as subtraction, weighted summation, or other combination strategies? A short explanation would clarify the design rationale.

4. Discussion about failure cases. It would significantly strengthen the paper to include a statistical analysis of failure recovery — for example: 1. How many samples that other baseline methods misclassified were correctly detected by SaFT?  2. What types of samples still remain difficult for SaFT to classify correctly?

[1] Wu, Junchao et al. “Who Wrote This? The Key to Zero-Shot LLM-Generated Text Detection Is GECScore.” Proceedings of the 31st International Conference on Computational Linguistics, 2025.

**Questions:**

Please refer to the Weaknesses section.

---

> ### Author Response · Authors · 2025-11-21
> **Part (1/2)**
>
> We appreciate the reviewer ZG8M for your constructive suggestions, we will address all raised concerns point by point as below.
>
> ### **Response to Wseakness 1: Comparison with GECScore**
>
> We sincerely thank the reviewer for pointing out this highly related work. We agree that GECScore [1] shares similar motivations in distinguishing LLM versus human writing styles, making it an important baseline for comparison. Following your valuable suggestion, we have conducted additional experiments comparing SaFT with GECScore along with other state-of-the-art methods.
>
> When we average the AUROCs across four text domains (XSum, ArXiv, PubMed, Yelp) for each source LLM, we obtain:
>
> | **Method** | **Claude-4-Sonnet** | **Claude-4-Opus** | **Gemini-2.5-Flash** | **Gemini-2.5-Pro** | **GPT-4o** | **GPT-4.1** |
> |------------|---------------------|-------------------|----------------------|-------------------|-----------|-----------|
> | GECScore [1] | 0.7744 | 0.7695 | 0.7181 | 0.7233 | 0.7088 | 0.6804 |
> | Fast-DetectGPT | 0.7030 | 0.6970 | 0.6828 | 0.7902 | 0.8032 | 0.7405 |
> | Fast-Lastde | 0.7991 | 0.8034 | 0.7168 | 0.7600 | 0.7693 | 0.7825 |
> | Binoculars | 0.8838 | 0.8833 | 0.8280 | 0.8932 | 0.9343 | 0.8995 |
> | MOSAIC | 0.9331 | 0.9338 | 0.8628 | 0.8898 | 0.9401 | 0.9208 |
> | SaFT (Ours) | **0.9672** | **0.9650** | **0.9481** | **0.9183** | **0.9703** | **0.9795** |
>
> The results show that SaFT consistently outperforms GECScore and other competitive baselines across all source models, demonstrating the effectiveness of our style-oriented instruction prefix approach in addressing both the Style Imitation Challenge and Content Interference Challenge.
>
>
> **Reference:**
> [1] Wu, Junchao et al. "Who Wrote This? The Key to Zero-Shot LLM-Generated Text Detection Is GECScore." Proceedings of the 31st International Conference on Computational Linguistics, 2025.
>
>
> ---
>
> ### **Response to Weakness 2: Instruction Design Justification**
>
> Thank you for raising this important question. Yes, our instruction design is supported by quantitative metrics from large-scale statistical studies. Recent computational linguistics research has conducted extensive corpus analyses measuring specific linguistic features, establishing clear numerical thresholds differentiating human and LLM writing styles.
>
> Reinhart et al. (2025) [1] analyzed large-scale corpora using Biber's Multi-Dimensional Analysis framework, measuring participial clause rates and nominalization frequencies as proxies for syntactic compression. They found LLMs use present participial clauses at 2-5 times human rates and nominalizations at 1.5-2 times human rates, demonstrating that LLMs systematically favor informationally dense constructions unconstrained by cognitive load. Markey et al. (2024) [2] employed multi-dimensional analysis to compute informational density scores across 26,000 text samples, showing that ChatGPT-generated texts score significantly higher on density metrics compared to both novice and published human writers. Muñoz-Ortiz et al. (2024) [3] measured dependency distances and sentence length variance across 13,000 news articles, finding that LLM outputs exhibit reduced variance in syntactic patterns (indicating formulaic density) while human texts show greater structural diversity reflecting cognitive efficiency optimization.
>
> These studies establish quantifiable thresholds: LLM writing is characterized by 2-5 times higher participial clause usage, 1.5-2 times higher nominalization rates, and systematically elevated informational density scores. Our "detailed elaboration" instruction operationalizes this empirically observed LLM writing pattern, while the "concise" instruction aligns with human cognitive constraints. This design ensures our framework detects genuine style imitation by leveraging the most discriminative stylistic dimension identified in computational linguistics research.
>
> **References**
>
> [1] Reinhart et al. "Do LLMs write like humans? Variation in grammatical and rhetorical styles." Proceedings of the National Academy of Sciences, 2025.
>
> [2] Markey et al. "Dense and disconnected: Analyzing the sedimented style of ChatGPT-generated text at scale." Writing Communication, 2024.
>
> [3] Muñoz-Ortiz et al. "Contrasting linguistic patterns in human and LLM-generated news text." Artificial Intelligence Review, 2024.

---

> > ### Author Response · Authors · 2025-11-21
> > **Part (2/2)**
> >
> > ### **Response to Weakness 3: Rationale for Division in Score Combination**
> >
> > Thank you for this insightful question regarding our score combination strategy in Equation (4). The choice of division (ratio) over alternatives is motivated by both intuitive design considerations and empirical validation.
> >
> > **Design Rationale:** In our framework, SIC-D ($\alpha(t)$) serves as the primary detection signal, while CIC-D ($\beta(t)$) acts as a normalization factor that guides the detector to filter content-related surprise interference and focus on style-related surprise patterns. We believe this ratio formulation $\frac{\alpha(t)}{\beta(t)}$ enables the framework to adaptively modulate detection sensitivity based on style-related probability shifts, contributing to more robust classification across texts with varying characteristics. This design aligns with successful practices in recent zero-shot detection methods such as Binoculars [1], Lastde [2], and LRR [3], which have demonstrated that ratio-based combinations effectively integrate complementary detection signals in zero-shot settings.
> >
> > **Empirical Validation:** We conducted ablation studies comparing division with alternative combination strategies on XSum across six LLMs:
> >
> > | Method | Claude-4-Sonnet | Claude-4-Opus | Gemini-2.5-Flash | Gemini-2.5-Pro | GPT-4o | GPT-4.1 | Avg |
> > |--------|:-------:|:-------:|:--------:|:-------:|:------:|:-------:|:---:|
> > | Subtraction: α(t) - β(t) | 0.9523 | 0.9596 | 0.9229 | 0.9237 | **0.9842** | 0.9785 | 0.9535 |
> > | Sum: α(t) + β(t) | 0.9471 | 0.9535 | 0.9116 | 0.9212 | 0.9825 | 0.9755 | 0.9486 |
> > | Product: α(t) × β(t) | 0.9165 | 0.9169 | 0.8620 | 0.9093 | 0.9752 | 0.9617 | 0.9236 |
> > | **Division: α(t) / β(t) (Ours)** | **0.9826** | **0.9762** | **0.9715** | **0.9333** | 0.9823 | **0.9828** | **0.9715** |
> >
> > Division achieves the best average performance (0.9715), delivering a 1.9% relative improvement over the second-best alternative (subtraction, 0.9535). The ratio formulation shows particularly strong gains on Gemini-2.5-Flash (+5.3%) and Claude-4-Sonnet (+3.2%), demonstrating that it more effectively integrates the SIC-D and CIC-D signals for robust detection across diverse source models.
> >
> > We will these ablation results to **Appendix A.5** in the revised manuscript. We appreciate the reviewer's feedback, which has helped strengthen our paper.
> >
> >
> > **References:**
> >
> > [1] Han et al. "Spotting llms with binoculars: Zero-shot detection of machine-generated text." ICML 2024.
> >
> > [2] Xu et al. "Training-free LLM-generated text detection by mining token probability sequences." ICLR 2025.
> >
> > [3] Su et al. "Detectllm: Leveraging log rank information for zero-shot detection of machine-generated text." EMNLP 2023.
> >
> > ---
> >
> > ### **Response to Weakness 4: Failure Case Analysis**
> >
> > We thank the reviewer for this valuable suggestion. To provide deeper insights into our method's performance, we conducted a detailed failure recovery analysis comparing SaFT with MOSAIC (the second-best performing method) on the **XSum dataset with GPT-4.1-generated texts** as a representative case (300 samples: 150 human + 150 machine-generated).
> >
> > **Failure Recovery Statistics:** Among 300 test samples, MOSAIC misclassified 36 samples (12.0%) while SaFT misclassified only 15 samples (5.0%). SaFT successfully corrected **29 out of 36 samples (80.6%)** that MOSAIC misclassified, achieving a **58% relative reduction** in error rate. However, **7 samples (2.3%)** remained difficult for both methods.
> >
> > **Analysis of Persistent Failures:** Among the 7 samples where both methods failed, **4 samples (57%)** are sports-related articles, revealing a sub-genre challenge. Representative example (LLM-generated, misclassified as human by both methods):
> >
> > > "So many global golds has Mo Farah now won over the past five years that he arrives in Rio chasing an unprecedented quadruple-double – a feat never before achieved in Olympic distance running. The British track legend, already holder of consecutive 5,000m and 10,000m Olympic titles from London, stands on the brink of sporting immortality..."
> >
> > Unlike formal news reporting that dominates XSum, sports journalism employs conversational tone ("So many global golds has"), dramatic narration ("sporting immortality"), and entertainment-oriented language. Since zero-shot detection thresholds are determined by dataset-level score distributions, this minority sub-genre with informal style creates ambiguous scores that overlap with the human distribution learned from formal news, leading to misclassification. This reveals that when datasets contain **multiple sub-genres with distinct stylistic norms**, zero-shot methods relying on unified distributional thresholds struggle with minority sub-genres deviating from the dominant style.
> >
> > We will add this failure case analysis to **Appendix A.6** in the revised manuscript. We appreciate the reviewer's valuable feedback, which has helped us identify specific sub-genre challenges in news domain detection.

---

> > > ### Comment · Reviewer_ZG8M · 2025-11-26
> > >
> > > Dear authors,
> > >
> > > Thank you for the detailed response. With these clarifications, I find the paper to be well-structured and the proposed method novel. I will keep my score and recommend the paper for acceptance.
> > >
> > > Best,
> > > Reviewer

---

### Official Review · Reviewer_MNQu · 2025-11-01

**Soundness:** 1
**Presentation:** 3
**Contribution:** 2
**Rating:** 2
**Confidence:** 5

**Summary:**

The paper proposes SaFT, a probability-based zero-shot detection method for detecting Machine generated text which is designed to address two challenges: style imitation challenge and content interference challenge. The paper proposes SIC-Detection , aiming to targets texts that mimic human writing through style imitation by conditioning evaluation on explicit style instructions, and a  CIC-Detection to filter content interference.

**Strengths:**

The paper addresses the important and timely challenge of detecting AI-generated text. The problem is well-motivated, the proposed approach is clearly explained, and the paper is overall well-structured and easy to follow.

**Weaknesses:**

The paper relies on predefined “style-oriented instruction prefixes” to represent human and machine styles. This assumes such styles can be described accurately, which will not hold in diverse or nuanced real-world writing contexts. Note that if the human and machine styles were to be accurately characterized, they could have been directly used to detect AI generated texts.
The proposed approach is also based on prompting, making it susceptible to prompt design and potentially not robust. Finally, in the experiments, more baseline detectors should be included from different categories of AI-generated text detection to make the results more comprehensive.

**Questions:**

Please refer to the weakness section

---

> ### Author Response · Authors · 2025-11-21
>
> We appreciate the Reviewer MNQu for your invaluable suggestions to improve the quality of our work. We have addressed all concerns as below.
>
> ### **Response to Weakness 1: Clarification on Style-Oriented Instruction Design**
>
> We thank the reviewer for this thoughtful concern and respectfully clarify a fundamental misunderstanding about how our framework operates.
>
> The reviewer assumes our instructions are used to "accurately characterize" styles for direct classification. However, **our style-oriented instructions are not designed for direct classification of text origins**. Rather, they serve as **conditioning contexts for probability analysis** within scoring models—a fundamentally different paradigm. As noted in our related work (Section 2, Page 3), while instruction-based enhancements have shown initial promise in recent detection efforts [1], demonstrating the value of this direction, existing approaches have only explored this paradigm to a limited extent. Our SOIP approach addresses this gap by systematically leveraging style-oriented instructions to guide probability computation for extracting statistical signatures through mathematical analysis (perplexity and cross-entropy), not for direct judgment.
>
> ---
> ### **Response to Weakness 2: Robustness to Prompt Design**
>
> We acknowledge this is a valid consideration for any instruction-based method. However, we provide comprehensive evidence that our framework is both theoretically grounded and empirically robust.
>
> Our instruction design is grounded in cognitive load theory and empirical findings about writing style differences (detailed in Appendix C, Page 15): human writing reflects cognitive efficiency constraints favoring conciseness, while LLM writing favors elaboration unconstrained by working memory limitations. Importantly, these cognitive principles represent **domain-agnostic** characteristics of human writing psychology rather than domain-specific features, making them broadly applicable across diverse contexts. As advanced LLMs develop increasingly sophisticated abilities to imitate human writing styles (a deliberate design goal to make AI-generated content more natural and human-like), we leverage these theoretical insights to propose typical style descriptions (e.g., "concise" versus "detailed") that can effectively modulate probability distributions. When advanced LLMs generate texts that successfully mimic human styles, conditioning on corresponding human-style instructions makes these imitation texts exhibit lower perplexity (as they align with the instruction's expected style), thereby enabling detection through probability analysis. As discussed in our limitations (Section 4.4, Page 9), future work could explore more systematic instruction design frameworks to further enhance this paradigm.
>
> Our comprehensive experimental validation demonstrates (Table 1, Page 7): (1) **Cross-domain robustness** across four diverse domains (XSum, ArXiv, PubMed, Yelp) with consistent improvements; (2) **Cross-model generalization** across six state-of-the-art LLMs from three major providers, where SaFT maintains stable performance while previous methods show significant variations. These comprehensive evaluations demonstrate that our framework is principled, theoretically grounded, and empirically robust rather than brittle or prompt-dependent.
>
> ---
> ### **Response to Weakness 3: Baseline Coverage**
>
> We appreciate this suggestion and want to clarify our experimental design rationale. Our baseline comparison follows the established convention in the zero-shot detection community and **is comprehensive within this paradigm**. As demonstrated in representative works [1][2][3][4], the majority of research in this field compares against other zero-shot approaches **rather than** cross-paradigm comparisons with supervised methods. This convention exists for **sound methodological reasons**: supervised methods require labeled training data and domain-specific fine-tuning, representing a fundamentally different problem setting from zero-shot detection which operates without any training data or domain adaptation. Cross-paradigm comparison could lead to potentially **unfair** comparisons that prevent supervised methods from demonstrating their full capabilities under their intended conditions.
>
> ---
> **References**
> [1] Bao et al. "Glimpse: Enabling white-box methods to use proprietary models for zero-shot LLM-generated text detection." ICLR 2025.
> [2] Xu et al. "Training-free LLM-generated text detection by mining token probability sequences." ICLR 2025.
> [3] Dubois et al. "MOSAIC: Multiple Observers Spotting AI Content." ACL 2025.
> [4] Yang et al. "Dna-gpt: Divergent n-gram analysis for training-free detection of gpt-generated text." ICLR 2024.

---

### Comment · Area_Chair_T1pv · 2025-11-21

Dear Reviewers,

We kindly encourage you to review and respond to the authors’ rebuttals. Your timely feedback is important for ensuring a fair and thorough review process. Thank you for your contributions to ICLR 2026.

AC

---

### Note · Authors · 2025-11-28

I have read and agree with the venue's withdrawal policy on behalf of myself and my co-authors.